# Conditions for replay of neuronal assemblies

Gaspar Cano[1], Richard Kempter[1,2,3]*

**1** Institute for Theoretical Biology, Department of Biology, Humboldt-Universität zu Berlin, Berlin, Germany, **2** Bernstein Center for Computational Neuroscience Berlin, Berlin, Germany, **3** Einstein Center for Neurosciences Berlin, Berlin, Germany

* r.kempter@biologie.hu-berlin.de

## Abstract

From cortical synfire chains to hippocampal replay, the idea that neural populations can be activated sequentially with precise spike timing is thought to be essential for several brain functions. It has been shown that neuronal sequences with weak feedforward connectivity can be replayed due to amplification via intra-assembly recurrent connections. However, the mechanisms behind this phenomenon are still unclear. Here, we simulate spiking networks with different excitatory and inhibitory connectivity and find that an exclusively excitatory network is sufficient for this amplification to occur. To explain the spiking network behavior, we introduce a population model of membrane-potential distributions, and we analytically describe how different connectivity structures determine replay speed, with weaker feedforward connectivity generating slower and wider pulses that can be sustained by recurrent connections. Pulse propagation is facilitated if the neuronal membrane time constant is large compared to the pulse width. Together, our simulations and analytical results predict the conditions for replay of neuronal assemblies.

## Author summary

In this work, we study how neural activity can propagate through a network of neurons. We therefore consider the activities of defined groups of neurons, which are also called assemblies. We are particularly interested in understanding how a previously learned sequence of activity patterns of assemblies can be replayed. To this aim, we combine large-scale numerical simulations with simpler analytical descriptions of neural dynamics. Our simulations show that, if feedforward connections across assemblies are weak, pulses of activity can be amplified by intra-assembly recurrent connections, allowing for sequence retrieval across a wide range of network structures and parameters. We introduce a new theoretical framework to study sequential activity, deriving conditions for sequence replay to succeed, and unveiling how replay speed depends on different network parameters. Crucially, we find that subthreshold membrane potential distributions are essential in determining the properties of the activity pulse and whether replay can

**Data availability statement:** Sample code to reproduce all simulations is available in the GitHub repository http://github.com/gaspar-c/replay-conditions.

**Funding:** This work was supported by the Deutsche Forschungsgemeinschaft (DFG, German Research Foundation, project-ID 327654276—SFB 1315 to RK). The funders had no role in study design, data collection and analysis, decision to publish, or preparation of the manuscript.

**Competing interests:** The authors have declared that no competing interests exist.

succeed. Our findings contribute to understanding the mechanisms of hippocampal replay in particular, which is important for memory consolidation, as well as the propagation of activity across feedforward neuronal circuits in general.

## Introduction

The notion that groups of neurons could form assemblies that fire together and that these assemblies could then fire in a specific order is old [1]. In fact, the idea that some neural populations are activated sequentially is ubiquitous in neuroscience, with precise neuronal spike timing thought to be essential for several brain functions.

In the cortex, neuronal recordings in behaving animals have repeatedly yielded evidence of such precise spike-timing sequences [2–5]. Thus, networks of layers of neurons connected in a feedforward manner—called synfire chains—have been extensively studied to understand how such sequences of neuronal activity could propagate in the cortex [6–14]. These modeling studies have found that, given enough feedforward connectivity across groups of neurons, neuronal networks are capable of propagating an activity pulse indefinitely with precise spike timing. Interestingly, it has also been shown that adding feedback connections from one layer to the preceding one stabilizes this propagation [15].

In the hippocampus, sequential activation of learned neuronal sequences is believed to be essential for memory consolidation [16–22]. This phenomenon, known as hippocampal replay, has been observed in the brain of many animals, including humans [23–29]. Several computational works have investigated the properties of hippocampal neuronal networks to successfully replay a learned sequence [30–34].

Computational studies have also focused on how the interplay between recurrent connections (within an assembly) and feedforward connections (across assemblies) could influence the stability and sequential activation of activity patterns. Such recurrent connections are expected to exist in the brain due to Hebbian plasticity mechanisms ("fire together, wire together") and would be required for pattern completion, a key function attributed to the hippocampus [16,17,20,22,35]. More recently, [34] showed that in a balanced excitatory-inhibitory (EI) network, recurrent connections within an assembly of neurons can facilitate sequential activation of assemblies. If feedforward connections are not strong enough to propagate a signal across assemblies of neurons, then adding recurrent connections at each layer/assembly can amplify the signal and allow activity propagation to succeed. In such cases, neurons in an assembly need only fire once, just as what typically happens in synfire chains, where there are no recurrent connections. The authors of [34] hypothesized that, for recurrent connections to aid in sequence retrieval, a 'balanced amplification' mechanism [36] would be necessary. Such a mechanism requires a significant amount of feedback inhibition to the excitatory assemblies, which their model possessed. However, it remains unclear whether feedback inhibition and balanced amplification are necessary for recurrent connections to facilitate sequence retrieval.

The goal of this work is to understand the mechanisms by which recurrent connections enable or facilitate the feedforward activation of sequences of neuronal assemblies. To this end, we first study spiking networks of leaky integrate-and-fire (LIF) neurons capable of retrieving an embedded sequence. Then, we develop a time-discrete population model of the membrane-potential distributions for each assembly. Our population model provides analytical estimations that describe our simulations well and can predict how the replay speed depends on network parameters, as well as the necessary connectivity a sequence must have to be replayed.

## Results

To explain how recurrent connections can amplify propagating activity pulses and enable sequential activation of neuronal assemblies, we subdivide the Results in four major parts: numerical simulations of spiking neurons in 'Sequence retrieval in spiking networks' and 'Minimal spiking model reveals key replay mechanisms', numerical simulations of a mean-field model in 'A time-discrete population model for replay', and an analytical framework in 'Population model predicts analytical conditions for replay'.

### Sequence retrieval in spiking networks

We start exploring the mechanisms underlying sequence retrieval by performing spiking simulations of LIF neuronal networks with an embedded sequence. To study the ability of a network to recall the embedded sequence, we first ensure that it is in a low-firing asynchronous-irregular (AI) state. Then, to elicit replay, all the neurons of the first assembly in the sequence are made to fire simultaneously.

Fig 1 summarizes the results obtained for spiking networks. We studied three types of spiking network models (Fig 1–A). In each network, we embed a sequence of 10 assemblies. This number is chosen because it is short enough to allow for quick simulation times, but long enough to approximate the asymptotic pulse behavior in very long sequences. We embedded only one sequence in a network, and we assume that having several different sequences does not change the results on replay if the number of sequences is well below the capacity of the network; memory capacity or competition between different sequences [37] is not the focus of this manuscript.

There are several features, assumptions, and definitions that apply to all the spiking networks that we studied: If the connectivity (feedforward across assemblies and/or recurrent within assemblies) is strong enough, an activity pulse successfully propagates throughout the entire sequence (example raster plots in Fig 1–B,C; summary in Fig 1–D); otherwise, if connectivity is too low, the pulse dies out and the sequence cannot be retrieved (Fig 1–D, gray regions). To quantify retrieval, we consider an assembly to be 'fully activated' if a propagating activity pulse causes at least 90% of its neurons to fire once. Accordingly, if an activity pulse generated at the first assembly in a sequence successfully propagates through all of its assemblies such that each of them becomes 'fully activated' in sequence order, down to the very last one, we say that the sequence has been successfully 'recalled'/'retrieved'/'replayed' (these terms are used interchangeably). If the activity pulse dies out before reaching the last assembly in the sequence, we say that the replay has failed. Moreover, neurons should not fire more than once, on average. Such cases, with bursting or exploding activity that can extend to the whole network, are considered failed events. When retrieval is successful, we measure the asymptotic pulse width and propagation speed (Fig 1–D, colored regions). The width is obtained by filtering an assembly's activity and fitting it to a Gaussian to calculate its full width at half maximum (FWHM), which is averaged over the last three assemblies. The speed is calculated as the inverse of the time between filtered activity peaks of consecutive assemblies, averaged over the last four assemblies (for details, see 'Quantifying replay in the spiking network' in 'Methods').

To illustrate the putative mechanisms underlying replay, we start with a standard example model network and then reduce the complexity in further models. The first network (Fig 1–A1) is similar to that of [34], with each assembly made up of excitatory and inhibitory cells. In the second network, we remove inhibitory cells from the assemblies and use them only to provide global inhibition to the excitatory population (Fig 1–A2). Finally, in the third network, inhibition is completely

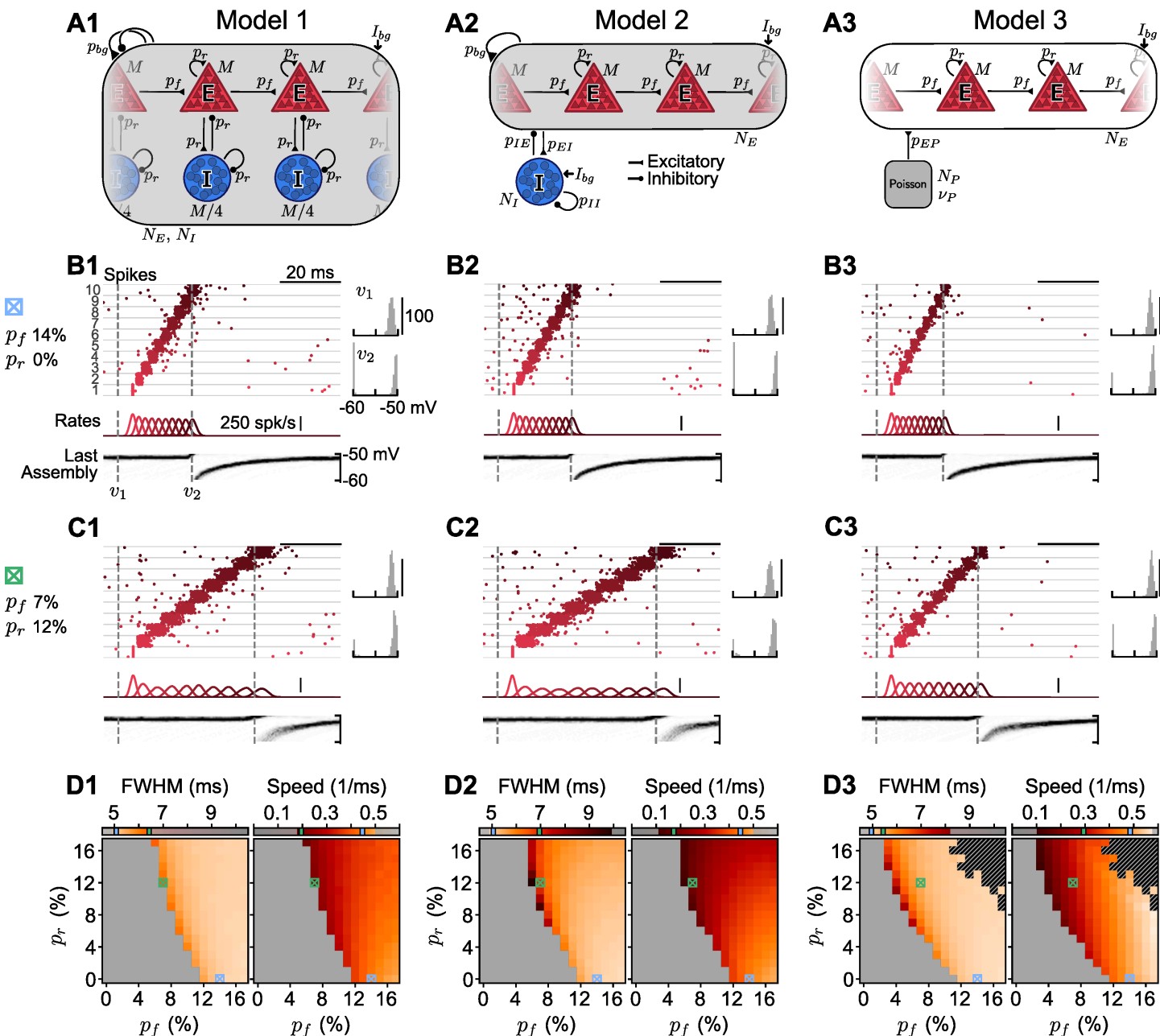

**Fig 1. Sequential activation of neuronal assemblies in spiking networks. A1–A3**, Network sketches of simulated spiking models. **A1**, Assemblies of $M = 500$ excitatory ($E$) and $M/4$ inhibitory ($I$) cells form a sequence embedded in a network with a total of $N_E = 20{,}000$ and $N_I = 5{,}000$ cells. Assembly cells form recurrent connections (probability $p_r$). The sequence of $q = 10$ assemblies is created by feedforward connections across the $E$ cells of each assembly (probability $p_f$). All neurons can form background connections (low probability $p_{bg} = 1\%$) and receive constant background current $I_{bg}$. The network is balanced by STDP in the $I \rightarrow E$ synapses until it reaches a low-firing AI state. **A2**, Similar to A1, but assemblies contain only $E$ cells, with inhibition being global. The same STDP mechanism is used. **A3**, Similar to A2, but without background connections ($p_{bg} = 0$) nor $I$ cells. $E$ cells receive Poisson inputs, tuned to replicate the AI state of the first two models. **B1–B3**, Simulations with $p_f = 14\%$ and $p_r = 0\%$. Networks are in an AI state when all the neurons in the first assembly are made to fire instantly. An activity pulse propagates throughout the sequence. 'Spikes', raster plot of sampled $E$ cells from each assembly (numbered 1–10 and separated by gray lines), in sequence order from bottom to top. 'Rates', assembly population rates (Gaussian-filtered with width 1 ms). 'Last assembly', membrane potential distribution of the last assembly (snapshots on the top right: $v_1$ and $v_2$). **C1–C3**, Same as B1–B3, but for $p_f = 7\%$ and $p_r = 12\%$. **D1–D3**, Asymptotic pulse 'FWHM' and 'Speed' of successful retrievals (unsuccessful ones shown in gray). Color bars have the same scale but only the measured ranges are colored in each panel. Crossed blue and green squares correspond to the parameters used in panels B and C, respectively, with lines in color bars indicating the measured values for each square. The black-hatched region in the upper-right corner of D3 represents failed retrievals due to activity explosions.

absent (Fig 1–A3). In the following, the three models and simulation results are described in more detail (see also 'Building the spiking network' in 'Methods' and Table 1 for a full list of network parameters).

**Excitatory-inhibitory assemblies.** Our first spiking model (Fig 1–A1) consists of $N_E = 20,000$ excitatory and $N_I = 5,000$ inhibitory cells, all of which form random background connections with low probability $p_{bg} = 1\%$. Then, a unidirectional sequence of $q = 10$ non-overlapping EI assemblies is embedded in the network. Each assembly consists of $M = 500$ excitatory and $M/4$ inhibitory cells that form additional recurrent connections to each other with some probability $p_r$. To make up the sequence, additional feedforward connections are created with some probability $p_f$ between the $M$ excitatory cells in subsequent assemblies.

All synapses are modeled as conductances that exhibit an instantaneous jump in synaptic conductance upon arrival of a presynaptic spike; thereafter, conductances decay exponentially. Furthermore, the model uses a spike-timing-dependent plasticity (STDP) mechanism in the magnitude of the conductance jump of $I \to E$ synapses [38], which balances the network until it reaches an AI state. Once the network is balanced, plasticity is turned off. Then, to investigate replay, we make all the neurons in the first assembly fire at once. If the generated activity pulse propagates throughout the sequence so that all sequence neurons fire approximately once in the correct order, the trial is considered successful. To study the role of recurrent connections in facilitating recall, this is attempted for different combinations of $p_f$ and $p_r$.

The example simulation in Fig 1–B1 indicates that recurrent connections are not always needed for replay. As shown in the summary figures in Fig 1–D1, $p_f$ must be high enough to retrieve the sequence, for any given $p_r$. Moreover, recurrent connections enable the retrieval of sequences with lower $p_f$, as in the example simulation in Fig 1–C1. The lower $p_f$ becomes, the higher $p_r$ is required, as in [34].

The simulations in Fig 1–B1,C1 show how the membrane potential distribution of the last assembly in the sequence behaves during the propagation of the activity pulse (panels at the Bottom and Right). We see that in the AI state, the subthreshold membrane potential distribution of each assembly approximates a normal distribution, with the upper tail of the Gaussian clipped at the threshold and causing the low firing rate driven by fluctuations (as predicted by existing theoretical models, [39,40]). When the propagating activity pulse reaches an assembly, this distribution is driven across the threshold until all neurons have fired. After firing, the membrane potential of each neuron is reset, and the distribution slowly settles again in the stationary AI state it had before the pulse disturbed it.

In the simulations in Fig 1–B1,C1, all neurons in the assemblies fire approximately once during the replay of the sequence; thus, moving the membrane-potential distribution across the threshold, without explicitly considering the reset and the refractoriness of individual neurons, is an appropriate simplification to understand the fast dynamics during replay, which we will use later in an analytical approach. The key difference in the behavior of the membrane potential distributions, as $p_r$ or $p_f$ are varied, is how long it takes the distribution to cross the threshold. If $p_f$ is high enough such that $p_r$ is not needed (as in Fig 1–B1), the distribution is fast to cross the threshold, causing all neurons to fire within a short time interval. As $p_f$ decreases below a critical value, the membrane potential distribution will take longer to cross the threshold (as in Fig 1–C1), because the feedforward synapses alone are no longer sufficient to make all neurons fire. Thus, recurrent excitatory synapses within an assembly need to cooperate with the feedforward synapses to ensure that all neurons fire, which means more time is needed for those synaptic inputs to arrive. The smaller $p_f$ is, the higher $p_r$ must be for recall to succeed, and the slower these events will be.

Accordingly, Fig 1–D1 demonstrates that, while all activity pulses start out as a perfectly synchronous activation in the first assembly generated by external stimulation, the asymptotic width and speed of the pulse depend on the chosen connectivity. In Fig 1–D1, we see that a larger width is correlated with a slower speed (see also S1 Appendix in Supporting Information). The FWHM varies between about 5 and 7 ms. It decreases with increasing $p_f$ and/or $p_r$, i.e., the higher the connectivity, the narrower the propagating pulse is (see 'FWHM' in Fig 1–D1). The panel 'Speed' in Fig 1–D1 shows a minimum pulse speed of ~6 ms per assembly (~0.18 assemblies/ms) and a maximum pulse speed of 2 ms per assembly

(0.50 assemblies/ms). Our maximum speed is roughly consistent with the excitatory synaptic model used, where synaptic conductances had a latency of 1 ms and an exponential decay time constant of 2 ms. The pulse speed increases considerably with increasing feedforward connectivity $p_f$, on which it seems to depend the most. However, there is a more complex dependence on $p_r$. In the region where $p_r$ is not needed for propagation ($p_f \geq 12\%$), increasing $p_r$ slows down the propagation speed. However, in the region where $p_r$ is needed ($p_f < 12\%$), increasing $p_r$ with respect to its minimal value slightly increases the pulse speed, although this effect appears to fade for increasing $p_r$. To investigate whether feedback inhibition and balanced amplification are essential mechanisms at play here, we look at two simpler spiking network models in Fig 1.

**Excitatory-assemblies balanced by global inhibition.** The second spiking network model that we simulate (Fig 1–A2) is as similar as possible to the first one, with the exception that the assemblies now contain only excitatory cells. Unlike the first model, the second one does not have specific assemblies of interneurons targeting each excitatory assembly through a higher probability of morphological connections. Instead, we have a homogeneous population of interneurons that has the same probability of receiving/sending synapses from/to any excitatory cell. A similar replay model has been studied in [33].

The simulations in Fig 1–B2:D2 show that the results obtained are nevertheless similar to those of the first model. Higher recurrent connectivity again allows for the retrieval of sequences with lower feedforward connectivity, and lower connectivity corresponds to slower and wider activity pulses: The observed pulse speed varies in a range similar to that of the first model, although it can reach lower values ($\sim$0.1 assemblies/ms). The minimal pulse width is also similar to the first model, but the maximum width can reach up to $\sim$10 ms. These slower and wider pulses likely occur because each assembly receives less feedback inhibition during replay compared to the first model.

Assembly membrane-potential distributions are similar to the first model in both the AI state and during retrieval. This is due to the use of the same STDP mechanism in all $I \rightarrow E$ connections present in the first model [38]. This mechanism allows inhibitory neurons to change their synaptic weights and target each neuron differently, which could lead to a selection of neurons based on the random $I \rightarrow E$ connectivity, thus somewhat mirroring the EI-assembly structure that was hard-wired in the first model. However, $E \rightarrow I$ and $I \rightarrow I$ connections, which do not possess any plasticity mechanism, are nonspecific and constant in the second model, which could cause the differences observed in pulse width and disrupt the EI coupling needed for balanced amplification to occur. In contrast, in the first model, a change in $p_r$ led to a change in all EI connections within an assembly.

In Model 1 and Model 2, we used an EI ratio of 4:1. Different ratios have been reported across brain regions (see, for example, [41]). Our choice of 4:1 was done for simplicity and is not critical to our results. In S2 Appendix (see Supporting Information), we show that the results shown in Fig 1–B2:D2 can be reproduced with different EI ratios, given an appropriate scaling of the synaptic inputs.

**Excitatory-assemblies only: A minimal spiking model.** The third spiking model (Fig 1–A3) is the simplest that can exhibit replay of assembly sequences. It consists merely of the sequence of excitatory assemblies and has neither inhibitory neurons nor background connections ($p_{bg} = 0$). This setup effectively disconnects the sequence from the background neurons, and we can consider a smaller network size $N_E = q \cdot M = 5,000$. Thus, neurons can connect only to those in the same assembly with probability $p_r$, or to those in the next assembly with probability $p_f$. The existence of recurrent connections distinguishes this minimal model from the 'synfire chain' architecture [7], where only feedforward connections across assemblies are considered. To keep the network in Fig 1–A3 in a low-firing AI state similar to the first two models, the constant background current $I_{bg}$ was decreased, and noisy excitatory synaptic inputs generated by Poisson processes were added [8]. $I_{bg}$ and the Poisson inputs were tuned such that the membrane potential distribution in the AI state was as similar as possible to the first two models.

The example simulations in Fig 1–B3:D3 show that successful replay events are similar to those of the first two models, with their pulse width and the speed also being in similar ranges. Furthermore, Fig 1–D3 demonstrates that weak feedforward connections can again be compensated for by high recurrent connectivity to allow for replay. However, a main

difference to the earlier models is that retrieval fails in regions where the total connectivity ($p_r + p_f$) is too high. These failed replay events (black and grey hatch in the upper right corners of the two panels of Fig 1–D3) correspond to bursting/exploding activity, which occurs because there is no feedback inhibition to prevent this from happening.

Together, the simulations in Fig 1–B3:D3 indicate that recurrent excitatory connections facilitate the sequential activation of assemblies even after completely removing inhibitory feedback from these networks. Although inhibition modulates the properties of propagating pulses and affects replay, our results suggest that inhibition-mediated balanced amplification is not needed for pulse propagation to be facilitated by intra-assembly recurrent connections, contrary to a previous conjecture [34]. Below, we simulate the minimal spiking model while varying further network parameters to elucidate critical features of replay.

### Minimal spiking model reveals key replay mechanisms

In what follows, we illustrate with numerical simulations the dependence of replay on recurrent and feedforward connectivity, synaptic latency, and membrane potential distributions in a minimal spiking network model with only excitatory neurons (sketch in Fig 1–A3). We again note that there are synaptic connections only within assemblies and across subsequent assemblies, but no background connections. These results allow us to speculate on which network mechanisms are responsible for replay success and failure, thus motivating the population model we introduce later in the manuscript.

**Dependence of replay on the synaptic delay.** The simulations of Fig 1 showed that the asymptotic speed and width of the activity pulse depend significantly on $p_r$ and $p_f$, with the lowest connectivities corresponding to the slowest and widest pulses. We expect that the speed of replay should also depend on the dynamics of excitatory synapses. To study this dependence, in Fig 2–A we simulate our minimal spiking model while varying the synaptic delay $\tau_l$ in the range 0.1– 4.0 ms, which was constant ($\tau_l = 1$ ms) in Fig 1. Note that in all the simulations discussed so far, we used exponentially decaying conductances to model our synapses. However, very similar results would be achieved with a different synaptic model, like an instantaneous delta current that increases the postsynaptic potential after some delay (see S3 Appendix in Supporting Information).

In each panel of Fig 2–A, we show the regions where replay succeeds (black and colored regions) and fails (gray), as a function of the connection probabilities $p_r$ and $p_f$. As expected, changes in synaptic delay significantly affect the speed of successful replay pulses, with the maximum measured speed decreasing from ∼1.42 assemblies/ms when $\tau_l = 0.1$ ms (not visible in the scale bars) to ∼0.21 assemblies/ms when $\tau_l = 4$ ms. Interestingly, the slowest speed for which replay can succeed is approximately the same in all panels of Fig 2–A (∼0.06–0.11 assemblies/ms), regardless of the synaptic delay $\tau_l$. Nevertheless, the connectivities $p_r$ and $p_f$ at which the minimal replay speed can be observed still strongly depend on the synaptic delay $\tau_l$. We can speculate on why this happens:

The combination of recurrent and feedforward connectivity, together with the speed of excitatory synapses, determines how much and how fast the subthreshold distribution of membrane potentials of a neuronal assembly moves across the firing threshold as an activity pulse propagates (insets in Fig 1–B,C). Replay can only succeed if the neurons' membrane potentials can go from their values in the AI steady state across the firing threshold. We hypothesize that, if this process is too slow, the generated activity is dissipated by the membrane leak currents, making it impossible for all neurons to cross and thus causing replay to fail. We note that the leak current is the only hyperpolarizing mechanism in our minimal model without inhibition, and its speed is governed by the membrane time constant $\tau_m$, which in all simulations was 20 ms. This idea would explain why the maximum speed changes as we change synaptic latency $\tau_l$, but the minimum speed remains approximately constant. Whether a pulse can be successfully replayed depends on whether the activation of each assembly is fast enough with respect to the leak currents. For any given combination of $p_r$ and $p_f$, the faster the synapses, the faster the activity pulse generated. Thus, with a slower $\tau_l$, we need stronger connectivity to reach the minimum required speed.

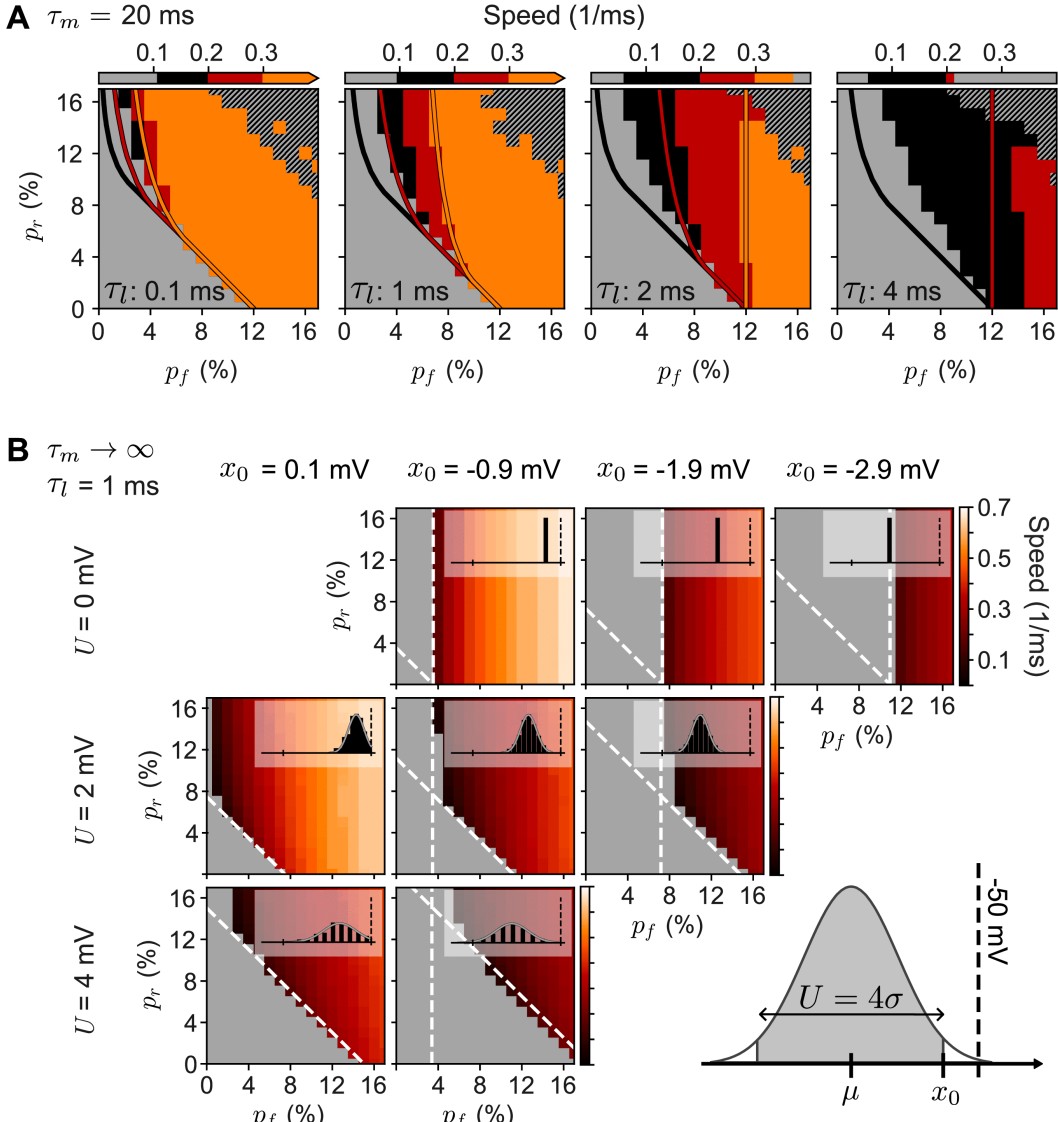

**Fig 2. Synaptic delay and distribution of membrane potentials condition sequence replay.** In all panels, we show the speed of replay as a function of connectivity. Results were obtained from spiking network simulations as in Fig 1–D3 (colored regions) and from analytical results (colored curves and dashed white lines). **A**, Speed of replay for different synaptic delays $\tau_l$ (indicated on the bottom of each panel). The simulations with $\tau_l = 1$ ms are also shown in Fig 1–D3. The black and colored regions represent successful replay; failed replays in gray. Orange regions correspond to propagation speed $\geq 0.3$ assemblies/ms; red regions to [0.2, 0.3) assemblies/ms; black regions to < 0.2 assemblies/ms. In the scale bars, the left edge of the black region marks the minimum speed for each panel, which varied between ~0.06 assemblies/ms (for $\tau_l = 4$ ms) and ~0.11 assemblies/ms (for $\tau_l = 0.1$ ms). The maximum speed varied between ~1.42 assemblies/ms (for $\tau_l = 0.1$ ms) and ~0.21 assemblies/ms (for $\tau_l = 4$ ms)—marked by the left edge of the grey region on the right of the scale bar (visible only on the third and fourth panels). The black-hatched regions in the upper-right corners (where $p_r + p_f$ is high) represent failed retrievals due to activity explosions. Simulation results are overlain by colored curves corresponding to the analytical predictions given by Condition 3 for three different values of speed: minimal speed measured in simulations (black curve); 0.2 assemblies/ms (red curve); and 0.3 assemblies/ms (orange curves). **B**, Speed of replay in a model without leak currents ($\tau_m \rightarrow \infty$), $\tau_l = 1$ ms, and fixed number of synaptic inputs per neuron. The firing threshold is located at −50 mV and the membrane potentials for each assembly before retrieval are drawn from a normal distribution with mean $\mu$ and SD $\sigma$, resulting in distributions (see insets) with different values of $U$ and $x_0$ (indicated to the left of and above each panel)—sketch on the bottom-right shows how $\mu$ and $\sigma$ relate to $U$ and $x_0$. We chose $U = 4\sigma$ and obtained a good match with our analytical Condition 1 (tilted dashed lines) and Condition 2 (vertical dashed lines)—for details, see Section 'Comparing spiking simulations to analytical conditions' in 'Methods'.

Together, the properties of the AI steady state membrane potential distribution of the assemblies are predicted to be essential for replay, in particular the width and the distance from the threshold, which are evaluated in more detail in Fig 2–B.

**Dependence of replay on the membrane potential distribution.** The membrane potential distributions of assemblies in the AI state of the simulations of the spiking network (Fig 1 and Fig 2–A) fit well to a Gaussian with mean $\mu \approx -51$ mV and SD $\sigma \approx 0.5$ mV (histograms in Fig 1–B). To study how the properties of the distribution of membrane potentials affect sequence replay, we systematically vary the width and the distance from the threshold in Fig 2–B. Furthermore, to simplify the interpretation of the results, we remove both Poisson inputs and neuronal leak currents from our minimal spiking model and, instead, manually draw each neuron's membrane potential from a clipped Gaussian distribution (inset at bottom right of Fig 2–B).

Fig 2–B confirms that both the mean and the spread of the membrane potential distribution significantly affect the connectivity necessary for replay to succeed. For example, when $p_r = 0\%$, there is a minimum value of $p_f$ below which replay is impossible, and this value increases with increasing distance between the lower edge of the distribution and the threshold. In other words, if there are no recurrent connections, the feedforward connections between assemblies must be strong enough to move the whole distribution across the threshold. Furthermore, increasing $p_r$ decreases the minimum $p_f$ required for the replay to succeed, which we also saw in the simulations in Fig 1–D and Fig 2–A. In other words, recurrent connections within assemblies can facilitate replay, but this amplification slows down replay. Finally, for most distributions, there is a minimum value for $p_f$ below which replay is impossible, regardless of how high $p_r$ is. This minimal $p_f$ depends on the distance between the right edge of the distribution and the threshold: If the feedforward connections are too weak to bring the upper edge of the distribution across the threshold, an assembly is not activated at all, and recurrent connections cannot help with replay. All of these qualitative insights will be quantified through analytical results in what follows.

### A time-discrete population model for replay

In this section, we model the dynamics of membrane potential distributions of neuronal assemblies during a replay event. To do this, we make several simplifying assumptions that are detailed below.

**Model assumptions.** Leak currents have been shown to be essential in determining subthreshold membrane potential distributions in spiking networks with AI firing [40], which is the baseline state in which we test replay in our simulations of Fig 1 and Fig 2. However, the histograms in Fig 1–B,C show that during successful retrieval events, the subthreshold membrane potential distribution retains its shape (the same shape as in the AI state) and is just moved toward larger voltages as synaptic inputs arrive. This observation motivates a separation of time scales: the slow time scale of the leak currents, given by the membrane time constant (20 ms in our model), and the fast time scale of sequence retrieval, governed by the time that excitatory synapses take to increase a neuron's membrane potential (1–5 ms). Thus, our first assumption is that we neglect the effect of leak currents during replay events. This assumption has two consequences for our population model. The first is that the membrane potential distribution has a fixed shape and only its mean can change (we also neglect the reset of the membrane voltage after it has reached the firing threshold, as explained below). The second consequence of neglecting leak currents (and reset) is that the mean of the membrane potential distribution can never decrease—it can only stay constant or increase. A limitation resulting from this assumption is that the model cannot describe the dynamics of very slow replays.

In principle, subthreshold membrane potentials in our spiking networks can have any value below the firing threshold. However, since each assembly has a finite number of neurons, their membrane potentials are found within a finite range of values. Moreover, our simulations show that in the low-firing AI state the distribution of membrane potentials is well condensed and close to the firing threshold (see histograms in Fig 1–B,C). Thus, for the population model, we assume

that each assembly's membrane potential distribution in the AI steady-state has a finite width $U$ that is identical for all sequence assemblies.

In our spiking simulations, a condition for successful replay is that each neuron only fires once during an event, on average, with bursting and exploding activity being considered failures to retrieve the sequence. In a successful replay event, after a neuron has fired, its membrane potential is set to the reset potential, and the neuron does not make any additional contribution (as seen in the histograms in Fig 1–B,C). Thus, we neglect the activity of neurons in an assembly once their membrane potentials have crossed the firing threshold. This assumption means that to evaluate successful retrieval in our population model, we only have to keep track of whether the distribution (which starts in the AI state) receives enough synaptic inputs to complete the crossing of the threshold once. From this assumption, it follows that the population model can never predict when activity explosions or bursting may occur.

During the replay seen in our spiking simulations, synaptic input is governed by recurrent connections within assemblies and feedforward connections across assemblies. Still, there may also be background connections (as in the first two spiking networks). In this case, the activity of an assembly can excite many other neurons in the network. However, this effect is negligible in our spiking models because the number of neurons in an assembly is not too large and the background connectivity is low (background connections are even absent in the third spiking model). Thus, we neglect background connections in the population model.

We have shown that, given the fast time scale at which excitatory synapses act on the sequence, the synaptic dynamics have a major impact on the speed of the emerging activity pulse and whether it can be replayed (see Fig 2–A). We further discussed that similar results are obtained whether synaptic currents are time-dependent conductances that decay exponentially (as in Fig 1 and Fig 2) or are instantaneous delta pulses (as in S3 Appendix — Supporting Information). This motivates our final assumption: If the synaptic dynamics is fast enough (with respect to the membrane time constant), the movement of the membrane potential distribution across the threshold during retrieval can be approximated by a time-discrete model in which the time step $\Delta t$ corresponds to the delay between a presynaptic spike and a postsynaptic potential.

**Model definition.** Given these assumptions, we model the distribution of membrane potentials $v$ of a given assembly $i \in \{1, 2, ..., q\}$ in a sequence of length $q$ at some time step $t \in \{0, 1, 2, ...\}$ with a static-shaped probability density function that has a finite width $U$ (example in Fig 2–B, bottom right). We define the threshold to be located at the potential $v = 0$ and consider that an assembly is active if at least a part of its distribution is above the firing threshold. Denoting $x_t^{(i)}$ as the position of the right edge of the distribution of assembly $i$ at time $t$, we say that this assembly is active at a given time if $x_t^{(i)} > 0$. Since $x_t^{(i)}$ cannot decrease with increasing $t$, the firing activity $a_t^{(i)}$ for some assembly $i$ at some time $t$ is the area under the distribution that crossed the firing threshold between the previous time $t{-}1$ and the current time $t$. Because the distribution is normalized, the total activity $\sum_{k=0}^{t} a_k^{(i)}$ is bounded between 0 and 1 and corresponds to the fraction of neurons in assembly $i$ that crossed the threshold up to some time t.

In our spiking network models, the initial condition is that all sequence neurons are in a low-firing AI state. This AI state is stable and does not cause sequential activation unless an external input is given to an assembly. In the population model, we capture this state by considering that all assemblies start at $t = 0$ with subthreshold membrane potential distributions at the same location and with zero activity, i.e.,

$$x_0^{(i)} = x_0 \leq 0 \,, \quad a_0^{(i)} = 0 \,, \quad \forall i \,. \tag{1}$$

For $t \geq 1$, in a given time step from $t{-}1$ to $t$, the right edge $x_t^{(i)}$ of the membrane potential distribution of assembly $i$ increases with respect to its previous position $x_{t-1}^{(i)}$ by some amount proportional to the input to that assembly. This input can come from recurrent synapses of the assembly itself with a weight denoted by $R$, from feedforward synapses of the previous assembly with weight $F$, or from some external input $J_t^{(i)}$, which we assume is applied to the entire assembly $i$.

These dynamics can be summarized as

$$x_t^{(i)} = x_{t-1}^{(i)} + R \cdot a_{t-1}^{(i)} + F \cdot a_{t-1}^{(i-1)} + J_t^{(i)} .$$

(2)

The first assembly in the sequence is $i = 1$, so, to keep Eq (2) consistent for the whole sequence, we formally define $a_t^{(0)} = 0$ for all $t$ (even though assembly $i = 0$ does not exist), and thus assembly 1 never receives a feedforward input. The connectivity weights $R$ and $F$ correspond to the average connection strength between neuronal assemblies, which are given by the average number of inputs (the product of the number of neurons in an assembly, $M$, with the connection probabilities $p_r$ or $p_f$) and the $E \rightarrow E$ synaptic weight $w^{EE}$ (how much one synaptic input increases the membrane potential of the postsynaptic neuron; see 'Comparing spiking simulations to analytical conditions' in 'Methods' for details), i.e.,

$$R = M \cdot p_r \cdot w^{EE}, \quad F = M \cdot p_f \cdot w^{EE} .$$

(3)

In our spiking simulations, we retrieved sequences by stimulating the first assembly. To best match this protocol in the population model, we consider external inputs that act only on the first assembly at $t = 1$, i.e.,

$$J_t^{(i)} = \begin{cases} J \geq 0 & , \quad \text{for } i = 1, t = 1 \\ 0 & , \quad \text{otherwise} . \end{cases}$$

(4)

In general, it is sufficient to have $J \leq U - x_0$, because a larger $J$ always fully activates the first assembly.

We consider that an assembly is successfully recalled if its distribution fully crosses the threshold, and we define $\bar{t}_i$ as the minimal time at which this happens for assembly $i$, such that $x_t^{(i)} \geq U$ for any $t \geq \bar{t}_i$. For a sequence with $q$ assemblies, we consider that the whole sequence has been successfully recalled if the membrane potential distribution of the $q$-th assembly fully crosses the threshold, i.e., if there exists some finite time $\bar{t}_q$ for which

$$x_{\bar{t}_q}^{(q)} \geq U .$$

(5)

We note that this definition of the successful recall of a whole sequence does not imply that all assemblies in the sequence are successfully recalled, which depends on initial conditions. However, we will show that this definition is useful, especially in the limit of large $q$.

In this work, we will describe the membrane-potential distribution in our time-discrete population model by two generic shapes, sketched in Fig 3. The first shape is a clipped Gaussian, which is chosen to best resemble the AI-state distributions observed in our spiking simulations (see Fig 1 and Fig 2–B). It consists of a Gaussian with standard deviation $\sigma = U/4$, which is then clipped at $\pm 2\sigma$ around the mean and normalized so that its integral is 1. The second shape is a rectangle with finite width $U$ and area 1, which is the simplest possible form a distribution could take. This approximation is chosen for its usefulness in analytical derivations, and we will refer to this particular case of the population model as the "rectangle model". In principle, the same approach could be applied to other shapes and, as we will see, some of the analytical results we derive from the population model are independent of the shape.

**Partial activation may lead to full activation via recurrent and feedforward connections.** The population model allows us to make predictions on the conditions for which a partial activation of the first assembly (as sketched in Fig 3) leads to the full activation of some subsequent assembly $i$. To derive this, we use the rectangle model, whose dynamics

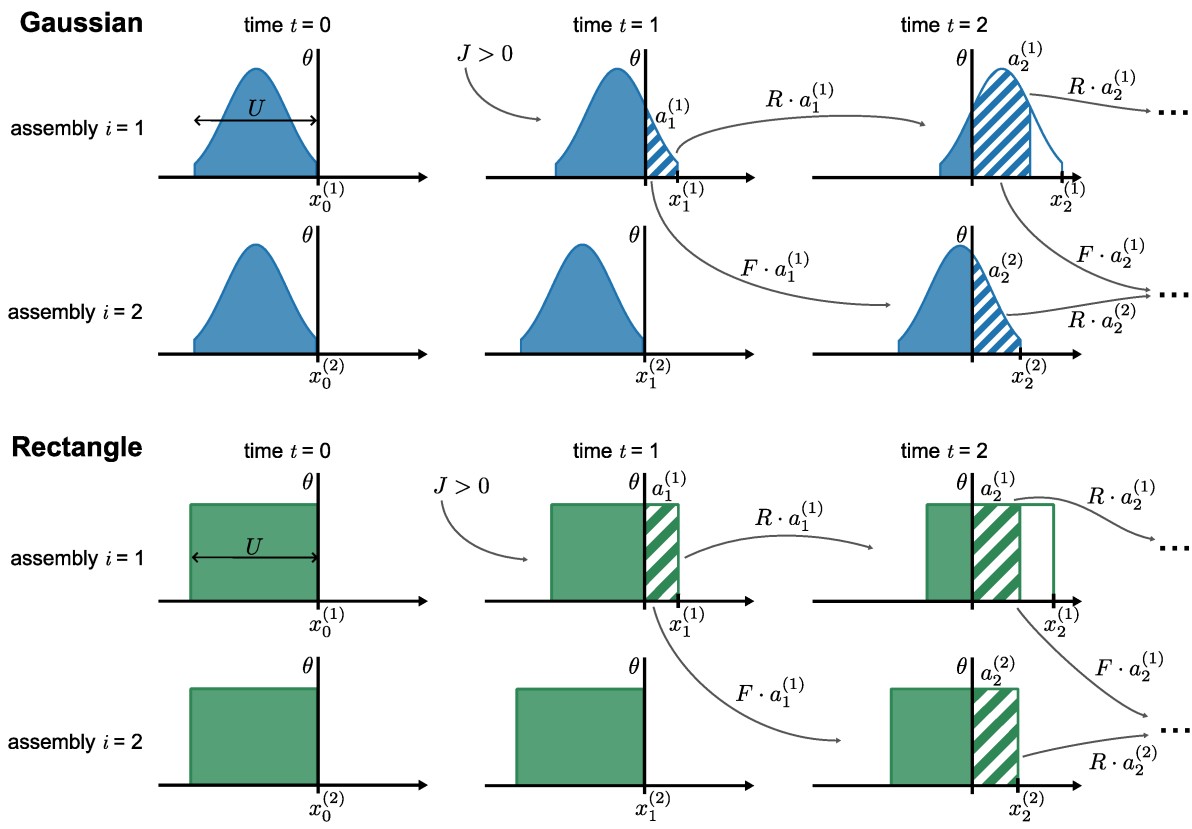

**Fig 3. Definition and dynamics of the time-discrete population model.** Sketch of the model's progression for the first two assemblies in a sequence at times $t = 0$, $t = 1$, and $t = 2$ for a clipped Gaussian (Top, blue) and rectangle (Bottom, green) distributions. Both assemblies start with initial condition (1) with $x_0 = 0$. At $t = 1$, the first assembly receives some positive external input $J>0$, generating some activity $a_1^{(1)}$. That activity (hatched areas) then propagates in the next time step $t = 2$ to that same assembly via recurrent connections ($R$) and to the next assembly via feedforward connections ($F$).

can be approximated (indicated by the "hat" accent over the symbol $x$) by the linear relation:

$$\hat{x}_t^{(i)} = \hat{x}_{t-1}^{(i)} + R \cdot \frac{\hat{x}_{t-1}^{(i)} - \hat{x}_{t-2}^{(i)}}{U} + F \cdot \frac{\hat{x}_{t-1}^{(i-1)} - \hat{x}_{t-2}^{(i-1)}}{U} , \tag{6}$$

which can be shown (for details, see the derivation of Eq (56) in 'Methods') to have the solution

$$\hat{x}_t^{(i)} = \begin{cases} 0 & \text{for} \quad t < i \\ J \cdot \left(\frac{F}{U}\right)^{i-1} \sum_{k=i}^{t} \binom{k-1}{i-1} \left(\frac{R}{U}\right)^{k-i} & \text{for} \quad t \geq i. \end{cases} \tag{7}$$

Let us define $m$ as the index of the first assembly to become fully activated at some time $\bar{t}_m$. Then, $m$ and $\bar{t}_m$ will correspond to the lowest $i$ and $t$ for which $\hat{x}_t^{(i)} \geq U$, i.e., for which the following condition is met:

$$J \cdot \left(\frac{F}{U}\right)^{i-1} \sum_{k=i}^{t} \binom{k-1}{i-1} \left(\frac{R}{U}\right)^{k-i} \geq U \tag{8}$$

If $i \leq m$ and $t \leq \bar{t}_m$, then $\hat{x}_t^{(i)} = x_t^{(i)}$ for the rectangle model, and Eq (8) is a necessary and sufficient condition for replay. Numerical simulations (see the next section) will show that this linear estimate provides a good approximation of the model dynamics for any distribution shape (and also when $i > m$ and $t > \bar{t}_m$), and thus can help explain the properties of successful replays in our spiking network simulations.

Before we turn to the comparison with numerical simulations, let us examine and discuss in more detail the linear estimate given by Eq (7), which already provides valuable insight into the dynamics of the population model. We therefore first consider the special cases $t < i$ and $t = i$: For $t < i$, we find $\hat{x}_t^{(i)} = 0$ because pulse propagation can never be faster than one assembly per time step. So, the right edge of the distribution of some assembly $i$ can only begin moving to the right for $t \geq i$. At $t = i$, assembly $i$ receives its first input, and $\hat{x}_i^{(i)} = J \cdot (F/U)^{i-1}$, which depends on the relative strength of only the feedforward connections $F$ with respect to the distribution width $U$. In each assembly $i>1$, the initial input $J$ is scaled by a factor of $F/U$, so that at assembly $i$ this scaling is $(F/U)^{i-1}$. Because recurrent inputs require additional time steps to arrive, they only contribute to an assembly's position for $t > i$. Thus, the dynamics of a sequence without recurrent connections (like a synfire chain) are fully described by what happens at $t = i$: If $F < U$, the initial input $J$ will be scaled down with increasing $i$ as the pulse propagates throughout the sequence, making replay impossible, regardless of the value of $J$. On the other hand, if $F > U$, the initial input is amplified as the pulse propagates, and, if the sequence is long enough, some assembly $m$ will eventually become fully activated once $J \cdot (F/U)^{m-1} \geq U$ is true.

If there are recurrent connections, for $t > i$ the initial feedforward input is scaled by an additional factor in Eq (7) that sums all the recurrent pathways (from $i$ to $t$) by which the initial input $J$ was scaled throughout the sequence, and the scaling factor depends on the relative strength of the recurrent connections $R$ with respect to the width $U$.

**Numerical simulations.** In Fig 4, we show simulations of the time-discrete population model for the clipped Gaussian distribution (Fig 4–A,B) and the rectangle distribution (Fig 4–C:E). For simplicity, we chose $U = 1$. To match the behavior of the AI state in the spiking model, where any input to an assembly leads to an increase in activity, we consider $x_0 = 0$. In the spiking network, we caused all neurons in the first assembly to fire at once, so in the population model simulations, we chose $J = U$ as the only input to the first assembly, which then becomes fully activated instantly. Fig 4–A:D indicates that clipped Gaussian and rectangle distributions produce similar results, both with the same qualitative behavior as the spiking network. The asymptotic pulse width and average speed (see 'Quantifying replay in the population model' in 'Methods'), as well as the ability to successfully replay the sequence, show a similar dependence on the strength of recurrent and feedforward connectivities; there are some minor differences between the two distributions for low feedforward connectivities. Finally, Fig 4–A,C shows that a large pulse width correlates with a slow pulse speed (quantification in S1 Appendix — see Supporting Information), which also resembles spiking simulations.

## Population model predicts analytical conditions for replay

To better understand the results of the numerical simulations, we will now describe analytical insights that can be obtained from the population model.

**Recurrent connections are not needed if feedforward connections are strong enough.** The numerical simulations in Fig 4–A:D (where $x_0 = 0$ and $U = 1$) show that recall always succeeds if $F \geq 1$, with those pulses propagating with minimal width at the maximum speed of one assembly per time step, and recurrent connections not being needed. It can be shown analytically that this must be true for any distribution of membrane potentials, regardless of its shape:

If a given assembly $m \geq 1$ becomes fully activated at some time $\bar{t}_m$ then we have $\sum_{k=0}^{\bar{t}_m} a_k^{(m)} = 1$. It follows that (see the derivation of Eq (30) in 'Methods' for details), if $F \geq U - x_0$, we have $x_{\bar{t}_m+1}^{(m+1)} \geq U$. In other words, assembly $m+1$ will become fully activated within one time step, regardless of the value of $R$. Consequently, the same applies to the next assembly $m+2$, which will become fully activated at time $\bar{t}_m+2$. Thus, we can infer that, if $F \geq U - x_0$ the full activation of an assembly will lead to the full activation of all assemblies downstream from it.

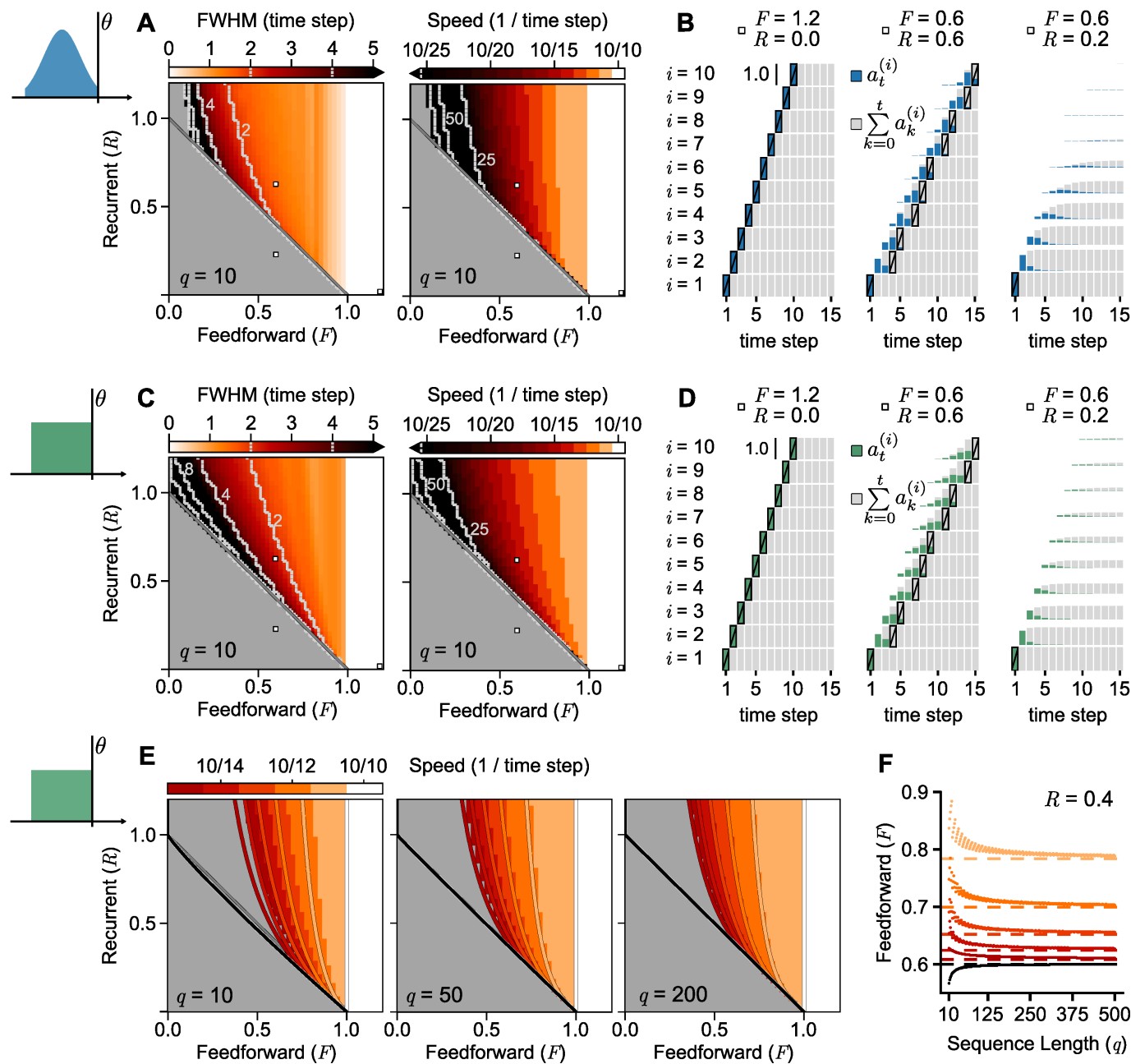

**Fig 4. Population model simulations confirm analytical predictions and match spiking network behavior. A**, Clipped-Gaussian model with $U = 4\sigma = 1$ and $x_0 = 0$, and external input $J = U$ in Eq (4). Retrieval succeeds if the last assembly in a sequence of $q = 10$ assemblies fully crosses the threshold at some time $\bar{t}_q$. Color maps: Asymptotic pulse width (FWHM in time steps) and approximate pulse speed ($q/\bar{t}_q$, i.e. retrieved assemblies per time step)—failed replays in gray—for different values of connectivities $R$ and $F$. Black regions show widths larger than 5 time steps and speeds slower than 10/25. The dashed white lines denote the upper borders for widths 16, 8, 4, and 2 (from left to right) and lower borders for speeds 10/100, 10/50, and 10/25. The three white squares correspond to the simulations on panel B. **B**, Example simulations; assemblies are stacked in sequence order from bottom to top. Colored bars represent activity $a_t^{(i)}$. Gray bars represent the sum of activities for that assembly. A black contour with a slash marks the time step at which an assembly becomes fully activated. **C–D**, Same as A–B, but for the rectangle model. **E**, Speed in rectangle model for different assembly sizes $q$. For $q = 10$, same panel as in B but the speed is capped at the minimum of 10/15 assemblies per time step. Overlain are colored lines, each corresponding to the $R$–$F$ values that border the condition (9) for a given pulse speed $S$. The black line corresponds to this analytical prediction when $S \to 0$. The $R + F = 1$ line is drawn in gray. The same is then shown for $q = 50$ and $q = 200$, with the speeds rounded up to match the same color levels as in $q = 10$. **F**, $F$-values of the analytical curves drawn in panel E when $R = 0.4$ (dots). Each dot is the minimal $F$ needed for replay to succeed, for the corresponding minimum speed, according to Eq (9). As $q$ increases, the needed $F$ converges to a fixed value, well approximated by Condition 3 (dashed lines).

**A necessary condition for sequence retrieval.** The simulations in Fig 4–A:D further show that the propagation of the pulse slows down as $F$ decreases, but recall can only succeed if $R + F > 1$. The closer $R + F$ is to 1, the slower the replay is. These results can also be shown analytically:

Without assuming any particular distribution shape, we can show (see the derivation of Eq (31) in 'Methods' for details) that

$$R + F > U - x_0 \qquad \text{(Condition 1)}$$

is a necessary condition for any assembly that does not receive external inputs (i.e., $i>1$ in our model) to be retrieved. For $x_0 = 0$ and $U = 1$, as in our numerical simulations, the condition becomes $R + F > 1$.

Condition 1 is necessary for replay for any distribution shape, but it is exactly sufficient only for the rectangle model (Fig 4–C; for an analytical derivation, see 'Rectangle model with $x_0 = 0$' in 'Methods'). The numerical simulations in Fig 4–A show that, although Condition 1 is not exactly sufficient for the clipped Gaussian (see failed regions of replay in the upper-left corners of 'FWHM' and 'Speed' panels), it nevertheless provides a good estimate of when replay will succeed. Note that sufficient conditions for replay are well approximated by Condition 1 only because our population model allows for extremely slow and wide replay pulses (black regions in Fig 4–A,C). Such pulses can propagate only if there are no leak currents.

Condition 1 can be applied successfully not only to the population model, but also to help us understand replay in simulations of spiking neurons. To illustrate the correspondence between theory and simulations, in the spiking networks simulated in Fig 2–B, we removed the membrane leakiness; in this case, the border between replay success and failure is well approximated by our analytical Condition 1 (oblique dashed white lines) for various positions $x_0$ and widths $U$ of clipped-Gaussian membrane potential distributions. As the population model predicts that Condition 1 is not only a good match when neglecting leak currents, but that it is always a *necessary* condition for replay to succeed, we should expect it to emerge in all our spiking simulations, regardless of the synaptic dynamics and membrane time constant used. Consequently, the need to fulfill Condition 1 can be observed in the simulations of Fig 1–D and Fig 2–A: there is always a straight oblique line $p_r + p_f = 12\%$ below which replay is impossible (in Section 'Comparing spiking simulations to analytical conditions' in 'Methods' we discuss how the specific value of the connectivity depends on network parameters).

**A minimum feedforward connectivity is needed for successful replay.** In the population model simulations of Fig 4, we chose membrane potential distributions with $x_0 = 0$ such that any feedforward input to an assembly would lead to the generation of some non-zero activity, as happens in the AI state of our spiking simulations of Fig 1. However, we could assume, without loss of generality, that an assembly $i$ starts with a distribution that has a gap below the threshold, such that $x_0^{(i)} < 0$. In a spiking network, that would correspond to having a distribution of membrane potentials fully sub-threshold, without spiking activity. We have simulated such a case in Fig 2–B. Although this does not capture the behavior of the AI state, it is useful to describe the activation of a subthreshold population through some feedforward input. This can also be motivated biologically: depending on how the excitatory-inhibitory connections are set in a network, an increase in activity in one assembly might cause hyperpolarization of other assemblies via lateral inhibition. Moreover, it has been shown that many hippocampal pyramidal cells become hyperpolarized during sharp wave-ripples [42–44], a phenomenon known to co-occur with hippocampal replay.

In the population model, for any activity to be generated at some assembly $i>1$, the sum of feedforward inputs from the previous assembly $i–1$ must be large enough to bring any portion of assembly $i$ over the threshold, i.e., the population model requires (see derivation of Eq (33) in 'Methods'):

$$F > -x_0 \qquad \text{(Condition 2)}$$

for any assembly that does not receive external inputs to be retrieved. If $x_0 = 0$ (as in our simulations of Fig 4), the condition becomes $F > 0$.

To compare this prediction to our spiking simulations of Fig 2–B, we follow the same methodology used for Condition 1 above (for details see 'Comparing spiking simulations to analytical conditions' in 'Methods'). We obtain a good match between Condition 2, given by the vertical dashed white lines in the panels of Fig 2–B, and the border between failed replay (gray region) and successful replays (colored region) in those spiking simulations. This again supports our population model as a good predictor of the behavior of spiking networks.

**Relation between pulse speed and connectivity.** In the context of the population model, we can approximate the pulse speed of successful replay pulses as $S = q/\bar{t}_q$, where $q$ is the sequence length and $\bar{t}_q$ the time step at which the last assembly becomes fully activated (for details see 'Quantifying replay in the population model' in 'Methods'). Thus, $S \in (0, 1]$, with the extreme cases $S = 1$ when $\bar{t}_q = q$ and $S \to 0$ as $\bar{t}_q \to \infty$ for a finite $q$. Notice that $S$ is measured in units 'assemblies per time step' and relates to the speed $s$ of the spiking model by $s = S/\Delta t$.

To find how connectivity determines the pulse speed, we will use Eq (7), which is the linear estimate obtained from the rectangle model. This estimate tells us that the replay can succeed with some speed $S$ if Eq (8) is true when $i = q$ and $t = q/S$, i.e.,

$$\frac{J}{U} \cdot \left(\frac{F}{U}\right)^{q-1} \sum_{k=q}^{q/S} \binom{k-1}{q-1} \left(\frac{R}{U}\right)^{k-q} \geq 1. \tag{9}$$

Importantly, the left-hand side of Eq (9) is increasing monotonically as $S$ decreases, so that if the condition is met for a certain speed $S'$, it will also be met for all speeds $S < S'$. Thus, a pulse speed estimate $\hat{S}$ can be obtained by fixing $R$ and $F$ and finding the largest $S$ for which Eq (9) is true. If this condition is not met even for $S \to 0$, then replay is impossible.

In the limit when $S$ is fixed and $q \to \infty$, it can be shown (see 'Rectangle model with $x_0 = 0$' in 'Methods') that Eq (9) becomes

$$F/U \geq \begin{cases} S \cdot \left(\frac{1-S}{R/U}\right)^{1/S-1} & \text{for} \quad R/U \geq 1 - S \\ 1 - R/U & \text{for} \quad R/U < 1 - S \end{cases}. \tag{Condition 3}$$

Remarkably, this condition is independent of the initial input $J > 0$—the initial condition is irrelevant for $q \to \infty$. Furthermore, Condition 3 also has a monotonous dependence on $S$, although this is not trivial. As with Eq (9), Condition 3 allows us to estimate the propagation speed $\hat{S}$ in dependence on the connectivities $R$ and $F$ by finding the largest $S$ for which the condition is true. Again, if this condition is not met even for $S \to 0$, then replay is impossible.

To illustrate our analytical results, in Fig 4–E we draw semi-numerically obtained curves that mark the border of Eq (9) for different values of the speed $S$: for various values of $F$ we pick the lowest $R$ that meets Eq (9); the resulting curve marks the connectivity needed for replay to reach that speed $S$. We first draw curves for $q = 10$ (Left panel) and vary the speed from its fastest value $S = 10/10$ (white vertical line) to its slowest $S \sim 10/\infty$ (black curve) and 5 finite values in between (colored curves). We also compare these estimates to the speeds measured in numerical simulations (white and colored regions, for $J = U = 1$, as in panels A:D). This comparison shows that the analytical curves slightly underestimate the needed connectivity for retrieval. An underestimation of the needed connectivity was expected, since our analytics were obtained from the linear estimate $\hat{x}_t^{(i)}$ of the location of the assembly, which always exceeds the true location $x_t^{(i)}$: Since $\hat{x}_t^{(i)} \geq x_t^{(i)}$, the connectivity needed to fully activate assembly $i$, i.e. $\hat{x}_t^{(i)} \geq U$, will be equal to or less than the connectivity needed for $x_t^{(i)} \geq U$.

In the next two panels in Fig 4–E, we see what happens to this profile in longer sequences with $q = 50$ (Middle) and $q = 200$ (Right). Although the curves given by the border of Eq (9) depend on $q$, we see that, as $q$ increases and $S$ is kept

fixed, both the numerical results and the analytical lines converge to a fixed location in the $R$–$F$ plane, which is approximately given by Condition 3 (see 'Comparing the linear estimate to rectangle-model simulations' in 'Methods' for why the numerical simulations also converge to Condition 3 as $q \to \infty$). To confirm this, in Fig 4–F, we fix $R = 0.4$ and plot the needed feedforward connectivity $F$ that meets Eq (9), while increasing the sequence length $q$. We see that, for the speeds considered in panel E, the connectivity needed to reach that speed (dots) converges to a fixed value, which is well approximated by Condition 3 (dashed line). Notice that in the limit when $S \to 0$ (black line), Condition 3 becomes $R+F \geq U$, which is the bound given by Condition 1 when $x_0 = 0$.

Let us once again turn to the application of analytical results on the population model to the results of simulations of networks of spiking neurons. Condition 3 shows how the recurrent and feedforward connectivity determine the replay speed $S$, measured in assemblies retrieved per synaptic time step $\Delta t$. If we want to understand how connectivity determines the replay speed $s$ as measured in the spiking network (in units of assemblies per ms), we need to consider the relation $S = s\,\Delta t$. Our spiking simulations in Fig 1–D and Fig 2–A showed that, for each network, there is a minimal propagation speed for which replay can succeed. We hypothesized that this happens because too-slow activity pulses are prevented from propagating by the leakiness of the membrane. If that is the case, then the minimum allowed replay speed in a spiking network is a constant $s_0$ that should not be strongly dependent on the synaptic dynamics. Given a constant $s_0 = S_0/\Delta t$, an increase in $\Delta t$ (i.e., having slower synapses) leads to a proportional increase in the minimum allowed speed $S_0$ in the population model. This, in turn, would lead to a stronger connectivity being required for replay to succeed. Consequently, a prediction of our population model is that increasing the duration of the synaptic time step $\Delta t$ should increase the connectivity required for replay to succeed. This prediction is confirmed by the simulations of Fig 2–A, where we saw how slower synapses required stronger connectivities.

We can also use the simulations of Fig 2–A to quantitatively test the prediction of Condition 3 on the replay speed in spiking networks (for details, see 'Comparing spiking simulations to analytical conditions' in 'Methods'). In each panel of Fig 2–A, we consider a different $\Delta t$, obtained by changing the synaptic latency $\tau_l$. For each of them, we plot curves corresponding to the success border of Condition 3 for three different values of replay speed $s$: the minimum speed measured for each case (black); 0.2 assemblies/ms (red); and 0.3 assemblies/ms (orange). We see that the analytical curves provide good matches for speeds 0.2–0.3 assemblies/ms, and a worse match for the minimal speed. This discrepancy is not surprising, since the population model from which Condition 3 emerges was derived under the assumption that leak currents can be neglected during replay. Thus, if, as hypothesized, these leaks are causing replay to fail in the spiking network, that border will not be well captured by our theory. On the other hand, if replay is fast enough such that the membrane leakiness can be neglected, our prediction seems to capture very well the relation between connectivity and pulse speed. Given all the assumptions and approximations needed to arrive at Condition 3, the fact that we can nevertheless obtain such a good approximation of replay speed in spiking networks shows how valuable our new population model approach can be.

To summarize, the three analytical conditions derived above provide accurate predictions of replay success and speed in our spiking network simulations, underscoring the validity of our population model in describing the replay of assembly sequences.

## Discussion

We considered the dynamics of sequential activation of neuronal assemblies by focusing on the cooperation between recurrent synapses within an assembly and feedforward synapses across assemblies. We simulated various spiking network models capable of recalling embedded sequences, showing that different connectivity schemes allow for sequential activation. Across models, we found that pulses propagated by weak excitatory feedforward connections between assemblies can be amplified by intra-assembly excitatory recurrent connections to retrieve neuronal sequences. We showed that excitatory synaptic dynamics and subthreshold membrane potential distributions are essential to determine whether this

Hebbian amplification can occur. Then, we introduce a time-discrete population model that provides analytical conditions that must be met for a sequence to be replayed. Our new theory describes our spiking network simulations and allows for an estimation of replay speed from network parameters.

To better frame the validity of our findings, we first summarize key model assumptions (and resulting limitations) used throughout this manuscript. Afterward, we discuss how our results fit within the existing literature on sequence replay.

## Main assumptions and limitations of our replay models

In spiking networks, we analyzed the role of recurrent and feedforward connections by varying their respective connection probabilities $p_r$ and $p_f$, within and between assemblies of $M$ cells (Fig 1 and Fig 2). Our population model (Fig 3 and Fig 4) assumes that a mean-field approximation of the dynamics of the spiking model is possible. Then we can define the strength of recurrent and feedforward connections as the product of the number of assembly neurons $M$, the connectivity $p$, and the postsynaptic potential $w^{EE}$; see Eq (3). For this approximation to hold, we require a sufficient number of synapses within and across assemblies ($M \cdot p \gg 1$) and that individual postsynaptic potentials $w^{EE}$ are not too large.

We further assume that in a successful replay event, sequence neurons need only fire once, on average. This allows us to neglect reset dynamics in our population model, which will therefore not be able to predict when activity explosions or bursts occur (see the top-right corner in Fig 1–D3). This means that our population model is only valid in a regime where runaway excitation is impossible, i.e., we are always in a subcritical regime where avalanches such as those studied in [45] cannot occur. How this might be ensured in neuronal networks is explained in more detail in the discussion of the role of inhibition below.

In all of our simulations of spiking networks, neurons start in a stable low-firing AI state when replay is elicited. We found that the distribution of membrane potentials in this baseline state is of key importance when determining the conditions for replay success; accordingly, the width $U$ of the distribution and its distance $x_0$ from the firing threshold enter the analytical Conditions 1, 2, and 3. The histograms in Fig 1–B,C show that, in our simulations of spiking networks, the distribution of membrane potentials is close to the firing threshold and is contained within a finite width. Theoretical conditions for such a state to arise have been formulated in [40], requiring that each neuron receives a large number of small synaptic inputs. In our first two spiking networks of Fig 1, this is ensured by having a large number of background inputs, i.e., $N \cdot p_{bg} \gg 1$. The AI state then emerges from the STDP rule of the inhibitory-to-excitatory connections introduced in [38]. Although these background connections are crucial for the baseline AI state, we assume that they are not essential for the dynamics during replay, because the background neurons remain with a very low firing rate. Thus, background connections are absent in our minimal spiking model and in the population model, in which appropriate membrane potential distributions were created by other means.

Our population model further assumes that an increase in the mean of an assembly's membrane potential distribution will generate activity in the network, proportional to the area of the distribution that crosses the firing threshold (see sketches in Fig 3). This assumption requires that there is a sufficient number $M$ of neurons per assembly, such that the distribution is well populated across the entire width $U$. Thus, our results are valid only if $M \gg 1$.

Another key assumption of our population model is that synapses are fast with respect to the membrane time constant. This allows us to neglect the effect of leak currents during replay events. This assumption is crucial to keeping the population model simple enough to allow for the analytical derivation of Condition 3, which we show to be an excellent predictor of replay speed in spiking networks for fast pulses (see the match between red and orange regions and lines in Fig 2–A). However, a drawback of this assumption is that our population model cannot predict when replay will fail because the activity pulse is too slow and is dissipated by the membrane leakiness (see the mismatch between the black region and lines in Fig 2–A).

Our minimal spiking model showed that feedback inhibition is not needed for the recurrent amplification of pulses. However, our spiking simulations show that inhibition modulates this amplification and affects the properties of the propagating activity pulse (compare different panels in Fig 1–C-D and see also S2 Appendix in Supporting Information). Thus, we expect that inhibition could play a significant role in retrieving sequences in biological networks. Although it is important to study these inhibitory effects, they are not captured by the population model introduced here, which, for simplicity, we have kept as simple as possible. Future work could focus on whether our new theoretical approach may also describe the effect of inhibitory currents during replay.

### The role of Hebbian assemblies in sequence replay

[46] introduced a recurrent network capable of storing multiple patterns of activity by strengthening auto-associative synapses with a Hebbian plasticity mechanism. However, the same study showed that this network was not robustly able to store and retrieve sequences of more than a few patterns. [47,48] later extended this to show that they could store discrete patterns of activity using Hopfield dynamics and embed them in a hetero-associative network with slower feedforward connections across patterns, thus being capable of reliably storing and retrieving sequences. These early theoretical studies consisted of networks of binary neurons, with each neuron having a 50% probability of being active for any given pattern. In such cases, patterns typically share many neurons, and the distinction between feedforward connections (across assemblies) and recurrent connections (within an assembly) is blurred. Many recent hippocampal models of sequential activity rely on continuous sequences of neurons, where distinct assemblies cannot be isolated at all [30,32,34,49]. However, other authors have hypothesized that auto- and hetero-association might be implemented by complementary but distinct subnetworks in the hippocampus [50].

A very different approach is the classical synfire chain model of sequential activity [7]. It consists of purely feedforward networks in which connections across distinct layers (or assemblies) of neurons are considered, but recurrent intra-assembly connections are typically disregarded. However, the abundant evidence on the existence of engram cell assemblies that encode external representations within recurrent networks (see [51] for a recent review) suggests that it is worth considering recurrent and feedforward connections within and across a sequence of assemblies. From a theoretical point of view, the most straightforward way of investigating this is to consider discrete non-overlapping assemblies [34,37], allowing for a clear distinction between recurrent and feedforward connections, which we have done here. Interestingly, a recent theoretical study [52], investigating how activity propagates throughout the brain, found that within-region and cross-region connectivity can play competing roles in the generation and transmission of neural signals.

Our simulations of spiking networks with balanced EI assemblies confirm that recurrent connections play a crucial role in maintaining the propagation of activity pulses, especially when the feedforward connectivity is too low, as proposed in [34]. The stronger the recurrent connectivity, the better the system is at amplifying the activity of assemblies, ensuring that activity pulses do not die out as they propagate across a sequence with too weak feedforward connections. The authors of [34] hypothesized that in balanced EI networks, increasing recurrent connectivity would lead to faster and stronger pulses through a 'balanced amplification' mechanism [36], thus requiring strong assembly-specific feedback inhibition. However, our simulations show that propagation speed is only weakly dependent on assembly-specific recurrent connections (Fig 1–D1). To assess the role of inhibition, we simulated two special kinds of spiking networks: one model where global, i.e. assembly-unspecific, inhibition balances excitation; and another model made up exclusively of excitatory assemblies without any inhibition, which we called 'minimal model'. These two networks produced similar amplification by excitatory recurrent synapses (Fig 1–D2-D3). This result suggests that balanced amplification is not the key mechanism at play when recurrent connections amplify weak activity pulses. Hebbian amplification alone, which is governed by intra-assembly recurrent excitatory connections, is sufficient and reliable in allowing for the retrieval of sequences when feedforward connectivity is too low.

In this work, we studied the replay of only unidirectional sequences of Hebbian assemblies. However, hippocampal replay, for example, is known to occur bidirectionally [23,28]. The results in [34] indicated that a configuration such as in our first model with EI assemblies (Fig 1–A1) is capable of replaying events bidirectionally if one embeds connections between assemblies in both directions. Because in our simulations the activation of one assembly ends due to the refractoriness following the activation of most of its neurons (Fig 1–B-C), we can infer that even in our minimal model without inhibition, such a bidirectional replay of sequences would be possible.

As we focus on the importance of Hebbian assemblies for sequence replay, an important question arises: how many neurons are there in a neuronal assembly? This number is the key parameter $M$ in our model. Crucially, our new theoretical approach allows us to make quantitative predictions for this value in the hippocampus, which is currently unknown. In Eq (3), we define recurrent and feedforward 'connectivity weights' as $R = M \cdot p_r \cdot w^{EE}$ and $F = M \cdot p_f \cdot w^{EE}$, respectively, where $p_r$ and $p_f$ are the connection probabilities, $w^{EE}$ is the synaptic strength, and $M$ is the number of neurons in an assembly. We then propose that replay can only succeed if $R+F > U$ (Condition 1, assuming for simplicity that $x_0 \approx 0$), where $U$ is the width of the membrane potential distribution. We can thus use $p$, $w^{EE}$, and $U$, which have been reported experimentally, to predict the value of $M$. Studies in CA3 indicate that $w^{EE} \approx 0.5$ mV [33], $p \approx 9\%$ [33], and $U \approx 10$ mV [53]. Thus, we expect that replay can only succeed if the number of neurons in a CA3 assembly is $M \gtrsim 100$.

### Replay and plasticity of synaptic connections

In our spiking network models, we embedded sequences by artificially creating synapses within and across assemblies. In most cases, synapses were created randomly, and for embedding a sequence, we created synapses with some probability that was greater than the overall background connectivity. In a biological system, this configuration is believed to emerge through changes in synaptic strength. Different types of plasticity rules have been proposed and found in the brain, from the classical Hebbian asymmetric temporal window of STDP in [54,55], to symmetric [56] or anti-Hebbian rules [57]. More recently, [58] used temporally asymmetric Hebbian learning rules to study the encoding and retrieval of random activity sequences in recurrently connected networks. This model was extended in [59,60] to show that the heterogeneity of plasticity rules allows for the control of the retrieval speed, with symmetric rules functioning as brakes and asymmetric rules as accelerators [60]. Other models of hippocampal replay have used symmetric [32] or asymmetric [49] STDP learning rules to encode neuronal sequences of place cells representing continuous space. In our work, we remained agnostic about the type of plasticity needed to embed our sequences and focused on the network properties needed to retrieve them.

### Recurrent and feedforward connections determine properties of the activity pulse

Both our spiking simulations and the population model show that, when the feedforward connectivity is strong enough, fast and narrow activity pulses can propagate without recurrent amplification, with most neurons within each assembly firing synchronously. This case is also observed in studies of cortical synfire chains [7–9]. Such studies also found that for a given feedforward connectivity, the failure or success of pulse propagation depends on how the sequence is initiated, i.e., the activity and width of the first pulse [8,9]. Moreover, the classical synfire chain literature predicts an all-or-none type of propagation across neuronal assemblies, with propagation either dying out or converging to a high and narrow activity pulse. In [34], it was shown that having EI assemblies allowing for recurrent amplification increases the range of activity and width of the first pulse so that feedforward propagation can be achieved. However, the asymptotic properties of these pulses were not investigated. Our work focused on finding these asymptotic features of the activity pulse and their dependence on the network properties. Thus, we did not investigate the dependence of pulse propagation on the properties of the initial pulse; instead, we always initiated sequences with perfectly synchronized pulses in the first assembly. We found that the relation between feedforward and recurrent connectivity shapes the asymptotic width and speed of sequential activity pulses (see Fig 1–D and Fig 4–A:D). For networks without recurrent connections, our model predicts

that successful pulses must eventually converge to fast and narrow-width propagation, as happens in classical synfire chains. As feedforward connectivity decreases below a critical value, we predict that successful pulses require amplification through recurrent synapses, needing more synaptic pathways to fully activate an assembly, and thus propagating at a slower speed and with a larger width, with neurons within an assembly firing less synchronously. Importantly, our population model allowed us to obtain an analytical estimate of how the asymptotic replay speed is determined by recurrent and feedforward connections. Our spiking simulations show that this estimation is good if replay is fast enough so that the membrane leakiness of a neuron can be neglected.

Interestingly, [61] found that, in a purely feedforward model, it is possible to decrease the synchrony of activity pulses in synfire chains by having a second sequence regulate the activity of the first one. Our findings suggest that recurrent connections among Hebbian assemblies can provide a different mechanism by which the transfer of wide (less synchronous) activity pulses can occur across neuronal circuits.

## Membrane potential distribution and leak currents condition retrieval success

The analytical Condition 1 derived from our population model shows that replay is possible only if the sum of the recurrent and feedforward synaptic weights exceeds the effective width of the distribution of the neuronal membrane potentials. This suggests that the properties of the membrane potential distribution of an assembly (especially the width $U$ but also its distance from the firing threshold $x_0$; see also Condition 2) are essential in determining whether a sequence can be retrieved or not. Moreover, our analytical estimate of the replay speed, given by Condition 3, also depends on the distribution width $U$. Thus, we propose that changing membrane potential distributions (via neuromodulation, for example) is a plausible mechanism by which replay success and its speed can be altered in the brain without the need for changing synaptic weights. This could be particularly advantageous given the wide range of hypothesized functions and circumstances that have been associated with replay—for example, sleep or awake replay, replay for consolidation or planning, etc. These observations are also in line with the modeling literature underlying the impact of noisy and balanced AI states in determining successful propagation in synfire chains [11,13,62].

Our model shows that, if all assembly neurons have a membrane potential distribution close to the threshold when the propagating activity pulse arrives, then Hebbian amplification by the recurrent excitatory synapses can be fast and reliable enough to allow for the activation of sequences where feedforward connectivity alone would not suffice. Balanced EI networks of the type studied by [34] possess such membrane potential distributions when they are in a low-firing AI state. However, such a state could also be achieved without inhibition by having noisy and uncorrelated external inputs to the sequence [8], as our minimal spiking model shows.

Another key insight from our study is that the membrane time constant can impose a limit on how slow an activity pulse can be to successfully propagate through a sequence of assemblies. Our findings suggest that, for a given membrane time constant, activity pulses relying on recurrent connections can propagate only if they are faster than some minimum speed. If the propagation speed of the pulse (determined by the synaptic delays and the interplay between recurrent and feedforward synapses) is too slow, then the neuronal leak currents governed by the membrane time constant dissipate the generated activity within an assembly before it can propagate to the next assembly. These observations allow us to infer how the membrane time constant influences sequence replay. Neurons with shorter membrane time constants may be more susceptible to disruptions in sequential activation, requiring stronger or faster feedforward synapses to propagate a highly synchronous pulse of activity. Therefore, having slower membrane time constants is an advantage in recalling sequences whose feedforward connectivity is too weak, but could nevertheless be retrieved by strong recurrent connections as a less synchronous pulse of activity. These predictions could be particularly interesting for understanding replay in the different regions of the hippocampus. Pyramidal neurons in the CA3 region have a significantly higher membrane

time constant (∼50 ms) than in the  (CA)1 region (∼10 ms) [63]. These long time constants could be useful in retrieving assemblies and sequences in CA3, a region known to have strong auto-associative connections [33,64–69]. Moreover, CA3 has been found to exhibit more robust replay than CA1 [70]. This observation supports the hypothesis that hippocampal sequences may first be learned and replayed in CA3 and then relayed to CA1 [71].

### Role of inhibition in replay

In the hippocampus, inhibition increases during replay events [72], with several studies showing that different interneuron types are involved in the mechanisms of replay-associated sharp wave-ripples [73–78]. [79] reported that inhibitory activity determines sequential spike times of pyramidal cells during sharp wave-ripples. However, it is unclear how important inhibition is for hippocampal replay.

Our results indicate how inhibition could affect replay. Importantly, the balance between excitation and inhibition determines the membrane potential distribution of neurons in an AI state. Our models show that the membrane potential distribution is a key component in replay dynamics. Appropriate membrane potential distributions were achieved by STDP of the inhibitory-to-excitatory connections [38] in our first two models and by manually tuned Poisson inputs for the third model (Fig 1). Furthermore, feedback inhibition can increase the parameter range for which retrieval is possible. For example, if the connectivity is high, the transient increase in activity observed during replay could cause bursting and activity explosions in the network, which could be prevented by feedback inhibition. For the range of connectivity studied in Fig 1, these explosions are only visible in the third model, where feedback inhibition is completely absent. However, if we were to increase $p_r$ or $p_f$ beyond what is shown, we would also see the first and second models creating these bursts (as was observed by [34] for high feedforward connectivity). Instead, if we fixed connection probabilities but increased the excitatory synaptic weights beyond a certain range, bursting and activity explosions would also occur (as seen in [33]). Moreover, if the synaptic weights were strong enough, a low-firing AI state would be impossible to reach without feedback inhibition in our minimal spiking model. As we aimed to understand when connectivity is strong enough for replay to succeed, we did not concern ourselves with the regions where replay fails because connectivity is too high. Finally, we note that such bursting or exploding activity could also be prevented in a network without inhibition by adding synaptic depression or spike-frequency adaptation mechanisms [33], which we have not considered.

To conclude, our combination of spiking simulations with an analytically tractable model allowed us to gain new insights into the mechanisms that regulate sequence retrieval. Our results contribute to the understanding of how neuronal networks can use recurrent connections within Hebbian assemblies to facilitate the activation of neuronal sequences.

## Methods

### Building the spiking network

We simulated the activity of various spiking networks and their ability to retrieve an embedded sequence of assemblies. Assemblies are subgroups of cells whose recurrent connections are, in most cases, considerably stronger than the average network connectivity. We also strengthened the feedforward connections across the embedded assemblies so that they form a sequence. This setup allowed us to study the conditions under which that sequence can be retrieved. In the following, we provide a detailed description of the structure of the networks and the tests we have performed on them.

Network simulations and analyses of the spiking network data were performed in Python (www.python.org), with the neural network being implemented with the package Brian [80], using a time step of 0.1 ms. The numerical values for all the parameters mentioned below are listed in Table 1 for three different models.

**Neuron and synapse models.** To keep our network models as simple as possible, neurons were described as LIF units. The subthreshold membrane potential $V_j(t)$ of a cell $j$ obeyed

$$C\frac{dV_j}{dt} = g^{\text{leak}}(V^{\text{rest}} - V_j) + I_j^{\text{syn}}(t) + I_{bg} \tag{10}$$

**Table 1. Parameter values for the spiking network models.** These parameters were used in the simulations shown in Fig 1 and Fig 2. In Fig 2–B, we set $g^{\text{leak}} = 0$ nS.

| Parameter | | Model 1 | Model 2 | Model 3 | Definition |
|---|---|---|---|---|---|
| $N_E$ | | 20,000 | 20,000 | 5,000 | Number of excitatory ($E$) cells |
| $N_I$ | | 5,000 | 5,000 | 0 | Number of inhibitory ($I$) cells |
| $M$ | | 500 | 500 | 500 | Number of $E$ cells in one assembly |
| $q$ | | 10 | 10 | 10 | Sequence length (# of assemblies) |
| $C$ | (pF) | 200 | 200 | 200 | Membrane capacitance |
| $g^{\text{leak}}$ | (nS) | 10 | 10 | 10 | Leak conductance |
| $V^{\text{rest}}$ | (mV) | −60 | −60 | −60 | Resting potential |
| $V^{\text{thr}}$ | (mV) | −50 | −50 | −50 | Voltage threshold |
| $\tau_{\text{refr}}$ | (ms) | 1 | 1 | 1 | Refractory period |
| $\tau_l$ | (ms) | 1 | 1 | 1 | Synaptic latency |
| $I_{bg}$ | (pA) | 200 | 200 | 58 | Constant background current |
| $p^{bg}$ | (%) | 1 | 1 | 0 | Background connectivity |
| $p^{IE}$ | (%) | - | 1 | - | (Global) $E$-to-$I$ connectivity |
| $p^{II}$ | (%) | - | 1 | - | (Global) $I$-to-$I$ connectivity |
| $p^{EI}$ | (%) | - | 1 | - | (Global) $I$-to-$E$ connectivity |
| $V^E$ | (mV) | 0 | 0 | 0 | Excitatory reversal potential |
| $g^{EE}$ | (nS) | 0.1 | 0.1 | 0.1 | $E$-to-$E$ synaptic weight |
| $g^{IE}$ | (nS) | 0.1 | 0.1 | - | $E$-to-$I$ synaptic weight |
| $\tau_d^E$ | (ms) | 2 | 2 | 2 | Excitatory synaptic decay time constant |
| $V^I$ | (mV) | −80 | −80 | - | Inhibitory reversal potential |
| $g^{II}$ | (nS) | 0.4 | 0.4 | - | $I$-to-$I$ synaptic weight |
| $g_0^{EI}$ | (nS) | 0.4 | 0.4 | - | $I$-to-$E$ initial synaptic weight |
| $\tau_d^I$ | (ms) | 4 | 4 | - | Inhibitory synaptic decay time constant |
| $\eta$ | | 0.01 | 0.01 | - | STDP learning rate |
| $\tau_{\text{STDP}}$ | (ms) | 20 | 20 | - | STDP time constant |
| $\nu_0$ | (spk/s) | 1 | 1 | - | STDP target firing rate |
| $N_P$ | | 0 | 0 | 5,000 | Number of Poisson units |
| $\nu_P$ | (spk/s) | - | - | 50 | Firing rate of Poisson units |
| $p^{EP}$ | (%) | - | - | 1 | $P$-to-$E$ connectivity |
| $w^{EP}$ | ($\mu$V) | - | - | 60 | $P$-to-$E$ synaptic weight |

where $C$ is the membrane capacitance, $g^{\text{leak}}$ is the leak conductance, $V^{\text{rest}}$ is the resting membrane potential, $I_j^{\text{syn}}(t)$ is the total synaptic current received by neuron $j$ at time $t$, and $I_{bg}$ is a constant background current injected into all neurons. Every time a neuron's membrane potential reaches the threshold $V^{\text{thr}}$, a spike is emitted and $V_j$ is reset to the reset potential (for simplicity, it equals $V^{\text{rest}}$), where it is clamped for a refractory period $\tau_{\text{refr}}$.

In the first two models (Fig 1, Left and Center panels), there were excitatory $E$ and inhibitory $I$ cells, and a conductance-based model was used for synapses such that

$$I_j^{\text{syn}}(t) = g_j^E(t)(V^E - V_j) + g_j^I(t)(V^I - V_j) \tag{11}$$

where $V^E$ and $V^I$ are the reversal potentials of excitation and inhibition, respectively. The time-dependent variables $g_j^E(t)$ and $g_j^I(t)$ describe the total synaptic conductances resulting from input to neuron $j$. The conductance dynamics was described by

$$\frac{dg_j^E}{dt} = -\frac{g_j^E}{\tau_d^E} + \sum_{k,f} g_{jk}^{JE} \delta(t - t_k^{(f)} - \tau_l)$$

$$\frac{dg_j^I}{dt} = -\frac{g_j^I}{\tau_d^I} + \sum_{k,f} g_{jk}^{JI} \delta(t - t_k^{(f)} - \tau_l), \qquad J \in \{E, I\}, \tag{12}$$

where $\delta(t - t_k^{(f)} - \tau_l)$ is the contribution of the $f$-th incoming spike from neuron $k$ at time $t_k^{(f)}$, with $\delta$ being the Dirac delta function and $\tau_l$ the latency between a presynaptic spike and the onset of the postsynaptic response. The quantities $g_{jk}^{JK}$ denote the unitary conductance increase resulting from a single spike in the presynaptic neuron $k$ of the population $K$ connected to the postsynaptic neuron $j$ of the population $J$, where $J, K \in \{E, I\}$ with $E$ corresponding to the excitatory synapses and $I$ to the inhibitory ones. Between presynaptic spikes, conductances decay exponentially with time constants $\tau_d^E$ or $\tau_d^I$.

In the third spiking network model (Fig 1, Right panels; Fig 2–A) there were only excitatory cells but no inhibitory cells, and neurons received additional synaptic input from an external population with $N_P$ units, with each unit generating spikes according to a Poisson process with a mean rate $\nu_P$. Synapses driven by the Poisson processes were modeled as delta currents such that the input to the cell $j$ is

$$I_j^{\text{syn}}(t) = g_j^E(t)(V^E - V_j) + C \sum_{p,f} w^{EP} \delta(t - t_p^{(f)} - \tau_l) \tag{13}$$

with $\delta$ being the Dirac delta function, $t_p^{(f)}$ being the $f$-th incoming spike from the Poisson unit $p$, and $\tau_l$ being the latency between a presynaptic spike and the postsynaptic response. The time-dependent variable $g_j^E(t)$ was described by the first line in Eq (12), while $w^{EP}$ denotes the unitary membrane potential increase resulting from a single Poisson-generated spike.

**Structure and connectivity.** The first spiking model (Fig 1, Left panels) had a total number of $N_E = 20,000$ excitatory and $N_I = 5,000$ inhibitory cells. All cells could form connections to each other with a low background probability of $p_{bg} = 1\%$. Then, 10 assemblies were formed by selecting groups of non-overlapping $M = 500$ excitatory and $M/4$ inhibitory cells and creating connections between them with some recurrent probability $p_r$. After the assemblies were formed, a unidirectional sequence was created by selecting the excitatory neurons in each assembly and forming connections to the excitatory neurons in the next assembly with some feedforward connection probability $p_f$. All $I \rightarrow E$ synapses were plastic (for details, see 'Balancing the network').

In the second model (Fig 1, Center panels), none of the $N_I$ inhibitory cells were part of the sequence. Each assembly was now made up exclusively of $M$ excitatory cells. Background connections, again with low probability $p_{bg}$, were also formed exclusively by excitatory cells. Instead, inhibitory cells had fixed connection probabilities to themselves and to/from the excitatory cells, and connectivity was independent of the values of $p_f$ or $p_r$. All inhibitory cells formed connections to each other with probability $p_{II}$, to the excitatory cells with probability $p_{EI}$, and received connections from all $N_E$ excitatory cells with probability $p_{IE}$. As in the first model, all $I \rightarrow E$ synapses were plastic (for details, see 'Balancing the network').

In the third model (Fig 1, Right panels; Fig 2–A), there were no inhibitory cells ($N_I = 0$) nor background connections ($p_{bg} = 0$). To keep the network in a low-firing AI state, all neurons received synapses from an external population $P$, which provided input current as described in Eq (13). Each of the $N_P$ units could form connections to each of the $N_E$ neurons with probability $p_{EP}$. There was no plasticity in the third model.

**Balancing the network.** In the first two models (Fig 1, Left and Center panels), the inhibitory-to-excitatory synaptic weights were subject to a plasticity mechanism that allows excitatory cells to receive different amounts of inhibition in order to balance the network, allowing it to reach an AI state [38]. Near-coincident pre- and postsynaptic firing potentiates the synapse, whereas lone presynaptic spikes depress it. To implement this plasticity rule, we assigned a synaptic trace variable $x_j$ to every neuron $j$, such that $x_j$ is incremented with each spike of that neuron and decays with a time constant $\tau_{\text{STDP}}$:

$$\begin{aligned} x_j \rightarrow x_j + 1, \quad &\text{if neuron } j \text{ fires,} \\ \tau_{\text{STDP}} \frac{dx_j}{dt} = -x_j, \quad &\text{otherwise.} \end{aligned} \tag{14}$$

The synaptic weight $g_{jk}^{EI}$ from inhibitory neuron $k$ to excitatory neuron $j$ was initialized with the same value $g_0^{EI}$ for all $I \rightarrow E$ synapses. The conductances were then updated at the times of pre-/postsynaptic events in the following manner:

$$g_{jk}^{EI} \rightarrow g_{jk}^{EI} + \eta g_0^{EI}(x_j - \alpha), \quad \text{for a presynaptic spike in neuron } k,$$
$$g_{jk}^{EI} \rightarrow g_{jk}^{EI} + \eta g_0^{EI} x_k, \quad \text{for a postsynaptic spike in neuron } j, \tag{15}$$

where $\eta$ is the learning-rate parameter, and the bias $\alpha = 2\nu_0 \tau_{\text{STDP}}$ is determined by the desired firing rate $\nu_0$ of excitatory postsynaptic neurons. In numerical simulations, the synapses are plastic for 5 s, after which the network reaches the anticipated AI state, and then plasticity is turned off. In S2 Appendix (see Supporting Information, we show that longer training times slightly change the mean and the width of the membrane potential distributions. For simplicity, we kept the short 5 s balancing time for all simulations in the main text (see [81] for a recent theoretical study on how different inhibitory plasticity rules can affect the replay of learned sequences).

In the third model (Fig 1, Right panels; Fig 2–A), we tuned the external Poisson processes and the constant background current to mimic the membrane potential distribution achieved by the plasticity mechanism used in the first two models; there, the resulting distribution fits well to a Gaussian with mean $\sim -51$ mV and SD $\sim 0.5$ mV. We therefore chose parameters $N_P$, $\nu_P$, $p^{EP}$, $w^{EP}$, and $I_{bg}$ ad hoc, such that the third model showed a similar membrane-potential distribution and a similar low-firing AI state in the middle of the explored parameter range of connectivity, i.e., for $p_f = p_r = 8\%$.

## Quantifying replay in the spiking network

To test whether a sequence can be retrieved, we first initialized the network with all its synaptic connections, including the embedded sequence that we wanted to retrieve. In the case of the first two models, we let the network run with STDP of the $I \rightarrow E$ synapses, in order to balance the activity of the neuronal populations. Once the network reached an AI state (which typically occurs after 5 s of simulation), the plasticity was turned off. In the third model, we waited 1 s from the start of the simulation to ensure that the initial conditions did not affect replay. Then, once the neurons in the sequence were in an AI state, all neurons in the first assembly were made to fire simultaneously by manually setting their membrane potential to the firing threshold. The population activity of each assembly was then filtered with a Gaussian kernel with SD 2 ms, and peaks of the filtered activity were detected using a script by [82].

For a retrieval event to be successful, there were three sets of requirements: First, the peaks must be larger than 30 spikes/s for each assembly, the peaks between consecutive assemblies must occur in the same order as imprinted by the feedforward connectivity, and the subsequent peaks must show some delay of at least $\tau_I$. Second, the filtered activity of each assembly is fitted to a Gaussian to find its standard deviation (SD); we then count the spikes within $\pm$ 3 SDs of the filtered peak and check that all assembly neurons fire approximately once without bursting (the number of assembly spikes must be within 90–110% of the number of neurons). Third, if there are background neurons (as in the first two models), their filtered population activity must stay below 30 spikes/s. Note: in the simulations of Fig 2–B, events were much slower and spread out in time. There, the following parameters were adjusted ad hoc to correctly detect replay success: population activities were filtered with a Gaussian kernel of 7 ms (instead of 2 ms), and the activity threshold was set to 15 spikes/s (instead of 30 spikes/s).

If a retrieval event was successful (the requirements above are met), we calculate the asymptotic properties of the activity pulse. The asymptotic speed of propagation was calculated by the inverse of the time difference in the peaks of consecutive assemblies, averaged over the last four assemblies. To obtain the asymptotic pulse width, we further calculate the FWHM for each assembly by multiplying the fitted SD by $2\sqrt{2\log 2}$, and average this across the last three assemblies in the sequence.

For each parameter configuration of a network, we performed five simulations (with 1-s intervals between simulations) on five different pseudo-random instantiations of the network. The fraction of successful retrievals (out of 25) gave us a percentage of success for each point in the parameter space. We considered a retrieval successful for a given parameter configuration if it succeeded in at least 80% of the attempted simulations. For all successful retrievals, we average our calculated values of asymptotic pulse speed and pulse width.

## Comparing spiking simulations to analytical conditions

In Fig 2, we compared spiking network simulations to the three analytical conditions derived from the population model. Here, we detail the assumptions and approximations needed to match these analytical predictions to our spiking network parameters.

In the spiking network simulations of Fig 1 and Fig 2–A, the membrane potential distributions of assemblies in the AI state fit well to a Gaussian with mean $\mu \approx -51$ mV and SD $\sigma \approx 0.5$ mV (histograms in Fig 1–B). In Fig 2–B, we sample these membrane potentials from normal distributions with different chosen values for $\mu \in \{-50.9, -51.9, -52.9\}$ mV and $\sigma \in \{0.0, 0.5, 1.0\}$ mV. To match our population model, we always assume $U = 4\sigma$, which encompasses most neurons within an assembly (95% of the Gaussian is within 2 SDs of the mean). From this assumption, it follows that the offset is $x_0 = \mu + 2\sigma - \theta$, where $\theta$ is the firing threshold, which is always located at –50 mV (for a sketch, see Fig 2–B, Bottom-Right). Thus, we get $x_0 = 0$ for our simulations of Fig 1 and Fig 2–A, and $x_0 \neq 0$ for the simulations in Fig 2–B.

When defining the population model, we relate the connectivities $R$ and $F$ to the connectivities $p_r$ and $p_f$ in the spiking model by Eq (3). This equation implies that each synapse leads to a constant increase of exactly $w^{EE}$ in the membrane potential of a postsynaptic neuron. Because we used conductance-based synapses, this increase actually depends on the membrane potential of the postsynaptic neuron. Moreover, the increase does not happen instantaneously since the injected currents are low-pass filtered by the membrane. Nevertheless, we estimate $w^{EE}$ by using the mean $\mu$ of the membrane potential distribution:

$$w^{EE} \approx g^{EE} \tau_d^E (V^E - \mu)/C \tag{16}$$

where $g^{EE}$ is the postsynaptic increase in conductance, $V^E$ is the reversal potential for excitatory synapses, and $C$ is the membrane capacitance. Given our model parameters (Table 1), we estimate $w^{EE} \approx -\mu \cdot 10^{-3}$ (note the minus sign indicating that even though $\mu$ is always negative, $w^{EE}$ is always positive).

Moreover, the relation given by Eq (3) assumes that each neuron in the spiking network receives exactly $M \cdot p_r$ inputs from the assembly it is in, and exactly $M \cdot p_f$ inputs from the previous assembly. However, this does not correspond to the connectivity matrix of the spiking networks simulated in Fig 1 and Fig 2–A; there, the connection between a neuron and another neuron in the same or the next assembly was determined as a Bernoulli trial with probability $p_r$ and $p_f$, respectively. Thus, in those networks, each neuron receives $M \cdot p_r$ recurrent inputs and $M \cdot p_f$ feedforward inputs *on average*, with SDs approximately given by $\sqrt{M \cdot p_r}$ and $\sqrt{M \cdot p_f}$. For simplicity, to test Condition 1 and Condition 2, in the simulations in Fig 2–B we fixed the in-degree of the connectivity matrix, ensuring that each neuron indeed receives *exactly $M \cdot p_r$* recurrent and $M \cdot p_f$ feedforward inputs.

Given the assumptions just described, we can rewrite Condition 1 and Condition 2 for sequence retrieval in the spiking network simulations of Fig 2–B as

$$p_r + p_f > u - \delta, \tag{17}$$
$$p_f > -\delta \tag{18}$$

where we introduced the normalized width $u$ of the membrane-potential distribution and its normalized distance $\delta$ from the threshold as

$$u := \frac{U}{M \cdot w^{EE}} \quad \text{and} \quad \delta := \frac{x_0}{M \cdot w^{EE}} \, . \tag{19}$$

To test Condition 3, we first note that the replay speed $s$ (in units of assemblies per millisecond) in the spiking networks is related to the speed $S \in (0, 1]$ in the population model by $s = S/\Delta t$, where $\Delta t$ is the population-model time step, which captures the duration of a synapse. In our spiking simulations, we used a conductance-based synaptic model for all excitatory synapses in which, after a delay of $\tau_l$, conductance first increases by a constant amount in the postsynaptic neuron and then decays exponentially with time constant $\tau_d^E$. Thus, there is no exact time point at which a synapse can be said to occur, and thus no exact value of $\Delta t$ can be chosen. For simplicity, we approximate $\Delta t$ as the time elapsed between the firing of a presynaptic neuron and the centroid of the increase in membrane potential in the postsynaptic neuron:

$$\Delta t \approx \tau_l + \frac{1}{1/\tau_m + 1/\tau_d^E} \tag{20}$$

where $\tau_m$ is the membrane time constant. Given our model parameters (Table 1), we obtain $\Delta t \approx \tau_l + 1.8$ ms.

Using Eq (19), we can then write Condition 3 in terms of the parameters defined for the spiking model:

$$p_f \geq \begin{cases} u \cdot s \, \Delta t \cdot \left( \frac{1 - s \, \Delta t}{p_r/u} \right)^{1/(s \, \Delta t) - 1} & \text{for} \quad p_r \geq u \cdot (1 - s \, \Delta t) \\ u - p_r & \text{for} \quad p_r < u \cdot (1 - s \, \Delta t) \end{cases} \tag{21}$$

Let us finally turn to an evaluation of the numerical value obtained for $u$. We again note that we could estimate $u$ from Eq (19) because we fixed the in-degree of the connectivity matrix and set the leak currents to zero, thus eliminating filtering of postsynaptic potentials by the membrane. Taking into account our fitted values for $\mu$ and $\sigma$ and the approximation for $w^{EE}$ in Eq (16), we derived $u \approx 8\%$. In the simulations of Fig 2–A, where we tested Condition 3, the estimate for $u$ is, however, larger; we found that $u = 12\%$ provides the best fit, which is most evident at $p_r = 0$ because there Eq (21) predicts $p_f \geq u$. We speculate that a more accurate prediction of $u$ would have to account for the variability in the number of inputs a neuron receives when the in-degree of the connectivity matrix is not fixed and for the fact that the inputs are filtered by the membrane. Because our aim was to test how well Condition 3 can predict the speed of successful replays, and for this the exact value of $u$ is not critical, we simply fixed $u = 12\%$ for the comparison of theory and simulations in Fig 2–A.

We will now provide a detailed description of the population model and the analytical solutions that we have derived from it. To facilitate reading, the text is intended to be self-contained. Thus, there is some repetition of what is written in 'Results'.

## Defining the population model

We model the distribution of membrane potentials $v$ of a given assembly $i \in \{1, 2, ..., q\}$ in a sequence of length $q$ at some time step $t \in \{0, 1, 2, ...\}$ with a static-shaped probability density function $z_t^{(i)}(v)$. The static shape has finite width $U$ and is described by a normalized function $h_U(v)$, where $h_U(v) \geq 0$ for $-U \leq v \leq 0$, and $h_U(v) = 0$ otherwise. In general, the function $h_U(v)$ can have any shape as long as these conditions are met. If we denote $x_t^{(i)}$ as the position of the right edge of the distribution $z_t^{(i)}$, we can write

$$z_t^{(i)}(v) = h_U \left( v - x_t^{(i)} \right) . \tag{22}$$

We define the firing threshold as located at the potential $v = 0$ and consider that an assembly is active if the membrane potential distribution crosses the firing threshold. Since $x_t^{(i)}$ cannot decrease with increasing $t$, the firing activity $a_t^{(i)}$ for some assembly $i$ at some time $t$ is the area under the distribution that crossed the firing threshold ($v > 0$) between the previous time $t-1$ and the current time $t$,

$$a_t^{(i)} = \int_0^{+\infty} \left[ z_t^{(i)}(v) - z_{t-1}^{(i)}(v) \right] dv, \quad \text{for} \quad t \geq 1. \tag{23}$$

Because the function $h_U$ is normalized, the total activity $\sum_k^t a_k^{(i)}$ is bounded between 0 and 1 and corresponds to the fraction of neurons in assembly $i$ that crossed the threshold up to some time $t$. We consider that all assemblies start at $t = 0$ with subthreshold membrane potential distributions at the same location and with zero activity, i.e.,

$$x_0^{(i)} = x_0 \leq 0, \quad a_0^{(i)} = 0, \quad \forall i. \tag{24}$$

For $t \geq 1$, in a given time step from $t-1$ to $t$, the right edge $x_t^{(i)}$ of the membrane potential distribution of assembly $i$ increases with respect to its previous position $x_{t-1}^{(i)}$ by some amount proportional to the input to that assembly. Said input can come from recurrent synapses from the assembly itself with a weight denoted by $R$, feedforward synapses from the previous assembly with weight $F$, or some external input $J_t^{(i)}$, which we assume is applied to the whole assembly $i$. These dynamics can be summarized as

$$x_t^{(i)} = x_{t-1}^{(i)} + R \cdot a_{t-1}^{(i)} + F \cdot a_{t-1}^{(i-1)} + J_t^{(i)}. \tag{25}$$

The first assembly in the sequence is $i = 1$, so to keep Eq (2) consistent throughout the sequence, we formally define $a_t^{(0)} = 0$ for all $t$, and thus assembly 1 does not receive feedforward input. Eq (25) can also be written in the explicit (non-recurrent) form

$$x_t^{(i)} = x_0 + R \cdot \sum_{j=0}^{t-1} a_j^{(i)} + F \cdot \sum_{k=0}^{t-1} a_k^{(i-1)} + \sum_{l=1}^{t} J_l^{(i)}, \quad \text{for } t \geq 1 \tag{26}$$

where we have used our initial conditions (24).

In our spiking simulations, we retrieve sequences by stimulating the first assembly. To best match this protocol in the population model, we consider external inputs that act only on the first assembly at $t = 1$, i.e.,

$$J_t^{(i)} = \begin{cases} J \geq 0 & \text{for } i = 1, t = 1 \\ 0 & \text{otherwise}. \end{cases} \tag{27}$$

In general, we assume $0 \leq J \leq U - x_0$.

## Quantifying replay in the population model

We consider that an assembly is successfully recalled if its distribution fully crosses the threshold, and we define $\bar{t}_i$ as the minimal time at which this happens for assembly $i$, formally $x_t^{(i)} \geq U$ for any $t \geq \bar{t}_i$. For a sequence with $q$ assemblies, we

consider that the whole sequence has been successfully recalled if the membrane potential distribution of the last assembly fully crosses the threshold, i.e., if there exists some finite time $\bar{t}_q$ for which

$$x_{\bar{t}_q}^{(q)} \geq U. \tag{28}$$

Analogously to the spiking models, in the population model we consider the pulse width in assembly $i$ to be the FWHM in time of its activity:

$$2\sqrt{2\log 2} \cdot \sqrt{\sum_k (k - \langle t \rangle_i)^2 \cdot a_k^{(i)}} \tag{29}$$

where the mean pulse time is $\langle t \rangle_i = \sum_k k \cdot a_k^{(i)}$. We assume that the FWHM is $2\sqrt{2\log 2}$ times the SD, as in normal distributions. To approximate an asymptotic pulse width, we average Eq (29) over the last three assemblies.

In the spiking network, we calculate the pulse speed of successful replays as the time difference between the activity peaks of consecutive assemblies, and then we average over the last four assemblies to get its asymptotic value. In the population model, we consider that the average pulse speed is approximately given by $S = q/\bar{t}_q$, i.e., how many assemblies were retrieved per time step. For short sequences, the initial condition (magnitude of input $J$) might affect the difference between $S$ and the asymptotic pulse speed. However, as the sequence length $q$ increases, the asymptotic pulse speed gets closer and closer to $S$.

## Population-model predictions for any distribution shape

**Recurrent connections are not needed if feedforward connections are strong enough.** If a given assembly $m \geq 1$ becomes fully activated at some time $\bar{t}_m$ then we have $\sum_{k=0}^{\bar{t}_m} a_k^{(m)} = 1$. According to Eq (26), the right edge of the next assembly $m+1$ at any time $t > \bar{t}_m$ is given by

$$x_t^{(m+1)} = x_0 + R \cdot \sum_{j=0}^{t-1} a_j^{(m+1)} + F \quad \text{for} \quad t > \bar{t}_m. \tag{30}$$

If $F \geq U - x_0$, we have $x_t^{(m+1)} \geq U$ already at $t = \bar{t}_m + 1$, regardless of the value of $R$, and assembly $m+1$ is fully activated at $\bar{t}_m + 1$. Consequently, the next assembly $m+2$ will become fully activated at time $\bar{t}_m + 2$, and so on. Thus, if $F \geq U - x_0$, the full activation of an assembly will lead to the full activation of all assemblies downstream from it, with a minimal width of zero time steps and a maximal propagation speed of $S = 1$ (one time step per assembly), regardless of the distribution shape $h_U$ or the value of $R$.

**A necessary condition for sequence retrieval.** We know that the sum of activities for any assembly is always smaller than or equal to 1, i.e., $\sum_{k=0}^{\infty} a_k^{(i)} \leq 1$ for any $i \geq 1$. We can then derive from Eq (26) that

$$R + F \geq U - x_0 \tag{31}$$

is a necessary condition for any assembly $i>1$ that does not receive external input to be retrieved at some time $\bar{t}_i$, i.e., $x_{\bar{t}_i} \geq U$. This condition is necessary for any distribution shape $h_U$.

**A minimum feedforward connectivity is needed for successful replay.** For any activity to be generated at some assembly $i>1$, the sum of feedforward input from the previous assembly $i-1$ must be large enough to bring some portion

of assembly $i$ above the threshold, i.e.,

$$x_0 + F \cdot \sum_{k=0}^{t-1} a_j^{(i-1)} > 0 .\tag{32}$$

Since $\sum_{k=0}^{\infty} a_k^{(i-1)} \leq 1$, Eq (32) implies that

$$F > -x_0 \tag{33}$$

is a necessary condition for any assembly that does not receive external inputs to be retrieved.

**Rectangle model with $x_0 = 0$**

Here, we define the rectangle model and detail the analytical relations that can be derived from it, considering the initial conditions (24) with $x_0 = 0$. We approximate the shape of the distribution $h_U$ by the simplest possible form it could take, i.e., a rectangle with finite width $U$ and area 1, such that

$$h_U(v) = \begin{cases} 1/U & \text{for } -U \leq v \leq 0 \\ 0 & \text{otherwise.} \end{cases} \tag{34}$$

The activity $a_t^{(i)}$ in Eq (23) generated by the rectangle distribution (34) is

$$a_t^{(i)} = \frac{\min\left(x_t^{(i)}, U\right) - \min\left(x_{t-1}^{(i)}, U\right)}{U} \tag{35}$$

because $x_{t-1}^{(i)} \geq 0$ and $x_t^{(i)} \geq 0$, resulting from our assumption $x_0 = 0$. If the assembly has not yet fully crossed the threshold at time $t$, the activity (35) is the simple linear relation

$$a_t^{(i)} = \frac{x_t^{(i)} - x_{t-1}^{(i)}}{U} \quad \text{for} \quad x_{t-1}^{(i)}, x_t^{(i)} \leq U .\tag{36}$$

In what follows, we first examine the rectangle model with $x_0 = 0$ in two simple cases: $i = 1$ and $R = 0$. Afterward, we look at how we can approximate a general solution for its dynamics.

**Activating the first assembly.** To determine the conditions for activating an isolated assembly, we start by looking at the behavior of the first assembly, $i = 1$, which does not receive feedforward input. We thus consider the dynamics of assembly 1, as given by Eq (25), when it receives some constant input $0 < J \leq U$ at $t = 1$,

$$x_1^{(1)} = J, \quad a_1^{(1)} = J/U .\tag{37}$$

If $J = U$, the first assembly becomes fully activated at $t = 1$, by definition. However, if $J < U$, we must determine what happens in the subsequent time steps. Following Eq (26), the right edge of assembly 1 is given by

$$x_t^{(1)} = \begin{cases} 0 & \text{if } t = 0 \\ R \sum_{k=1}^{t} a_{k-1}^{(1)} + J & \text{if } t \geq 1 \end{cases} \tag{38}$$

since the only external input is $J$ at $t = 1$, and there is no feedforward input. Moreover, we also know that at any time $t$ for which an assembly has not yet been fully activated, its activity is given by the linear relation in Eq (36). Using Eq (36) and Eq (38), we see that for any $t>1$ at which assembly 1 has not yet become fully activated, i.e., $x_t^{(1)} \leq U$:

$$a_t^{(1)} = \frac{x_t^{(1)} - x_{t-1}^{(1)}}{U} = \frac{R}{U} \cdot a_{t-1}^{(1)} \tag{39}$$

We can solve this equation with the initial condition $a_1^{(1)} = J/U$ stated in Eq (37) to find

$$a_t^{(1)} = \frac{J}{U} \cdot \left(\frac{R}{U}\right)^{t-1} \quad \text{for} \quad x_t^{(1)} \leq U. \tag{40}$$

Inserting Eqs (24) and (40) into Eq (38), we find that, while $x_t^{(1)} \leq U$:

$$x_t^{(1)} = \begin{cases} 0 & \text{if} \quad t = 0 \\ J \cdot \sum_{k=1}^{t} \left(\frac{R}{U}\right)^{k-1} & \text{if} \quad t \geq 1 \end{cases} \tag{41}$$

Our goal is to find whether assembly 1 will eventually become fully activated if given enough time steps, i.e.: Is there some finite time $\bar{t}_1$ for which $x_{\bar{t}_1}^{(1)} \geq U$? To determine this, we start by noting that:

$$J \cdot \sum_{k=1}^{t} \left(\frac{R}{U}\right)^{k-1} = \begin{cases} \frac{J}{1-R/U} \cdot [1 - (R/U)^t] & \text{for} \quad R \neq U \\ J \cdot t & \text{for} \quad R = U \end{cases} \tag{42}$$

If $R \geq U$, we see that $x_t^{(1)}$ grows at a constant or increasing rate, and assembly 1 will fully cross the threshold if given enough time steps. If $R < U$, $x_t^{(1)}$ is a monotonically increasing function of $t$ that asymptotically tends to the limit $J/(1-R/U)$ as $t \to \infty$. Thus, given enough time steps, assembly 1 will become fully activated if $J/(1-R/U) > U \Leftrightarrow J + R > U$—note that once $x_t^{(1)} > U$ the dynamics of Eq (41) no longer apply, but we can nevertheless know whether this will happen at some point. If $J + R = U$, the assembly will never fully activate within a finite number of time steps.

**Sequence retrieval without recurrent connections.** Let us next consider a sequence with multiple assemblies but without recurrent connections ($R = 0, F > 0$). Given the input in Eq (27), we have $x_1^{(1)} = J$ and assemblies only move once to the right, which occurs when the feedforward input arrives at time $t = i$. Thus, assembly 1 will generate positive activity only at $t = 1$ : $a_1^{(1)} = J/U$, which is the only input assembly 2 will receive.

Let us first consider the case $F < U$. The dynamics of assembly 2 will be

$$x_2^{(2)} = J \cdot \frac{F}{U} \quad \text{and} \quad a_2^{(2)} = J \cdot \frac{F}{U^2} . \tag{43}$$

Thus, assembly 3 will only move to the right at time $t = 3$, with

$$x_3^{(3)} = J \cdot \frac{F^2}{U^2} \quad \text{and} \quad a_3^{(3)} = J \cdot \frac{F^2}{U^3} . \tag{44}$$

If we continue iteratively, we find that, in general:

$$x_i^{(i)} = J \cdot \left(\frac{F}{U}\right)^{i-1} \quad \text{for} \quad F < U. \tag{45}$$

Thus, we see that for $F < U$, as the generated pulse moves through the sequence, it becomes weaker and weaker, and the assemblies become progressively less activated. Thus, retrieval is not possible for a sequence of any length $q \geq 2$, even if we fully activate the first assembly by setting $J = U$.

Let us now consider the case where $F \geq U$. We can derive for the second assembly that

$$x_2^{(2)} = J \cdot \frac{F}{U}, \quad a_2^{(2)} = \begin{cases} J \cdot \frac{F}{U^2} & \text{if} \quad J \cdot F/U^2 < 1, \\ 1 & \text{otherwise.} \end{cases} \tag{46}$$

Subsequently, $a_2^{(2)}$ is the only input assembly 3 will receive, and we can derive that

$$x_3^{(3)} = \begin{cases} J \cdot \left(\frac{F}{U}\right)^2 & \text{if} \quad J \cdot F/U^2 < 1, \\ F & \text{otherwise,} \end{cases} \tag{47}$$

$$a_3^{(3)} = \begin{cases} J \cdot \frac{F^2}{U^3} & \text{if} \quad J \cdot F/U^2 < 1 \wedge J \cdot F^2/U^3 < 1, \\ 1 & \text{otherwise.} \end{cases} \tag{48}$$

Since $F \geq U$, the condition $J \cdot F/U^2 < 1 \wedge J \cdot F^2/U^3 < 1$ is simply $J \cdot F^2/U^3 < 1$. If we continue iteratively, we find that, in general, for $i \geq 2$:

$$x_i^{(i)} = \begin{cases} J \cdot \left(\frac{F}{U}\right)^{i-1} & \text{if} \quad J \cdot F^{i-2}/U^{i-1} < 1, \\ F & \text{otherwise.} \end{cases} \tag{49}$$

Thus, the pulse will become stronger and stronger and assemblies become progressively more activated until one of them eventually fully crosses the threshold when the condition $J \cdot F^{i-1}/U^i \geq 1$ is met. After that, all downstream assemblies will also become fully activated, and the rest of the sequence is retrieved at the speed of one assembly per time step (since $F \geq U$). Thus, for $F \geq U$, recurrent connections are not needed for sequence retrieval.

### A linear estimate of rectangle-model dynamics

We now look at how a linear estimate of the rectangle model with $x_0 = 0$ can help us derive a general solution of the dynamics.

**Partial activation may lead to full activation via recurrent and feedforward connections.** If an assembly $i$ has not yet fully crossed the threshold at some time $t$, the activity in Eq (23) is given by the simple linear relation in Eq (36). Let us assume that $J < U$, so that assembly 1 does not become fully activated instantly. If the connectivity is strong enough, some assembly $m$ will be the first assembly to become fully activated, and we call the time this happens $\bar{t}_m$. Thus, any assembly $i \leq m$ at any time $t \leq \bar{t}_m$ will obey $x_t^{(i)} \leq U$, which allows us to use the linear relation in Eq (36), such that Eq (25)

becomes the following linear equation:

$$\hat{x}_t^{(i)} = \hat{x}_{t-1}^{(i)} + R \cdot \frac{\hat{x}_{t-1}^{(i)} - \hat{x}_{t-2}^{(i)}}{U} + F \cdot \frac{\hat{x}_{t-1}^{(i-1)} - \hat{x}_{t-2}^{(i-1)}}{U} \, . \tag{50}$$

As we want to generalize this equation to any assembly at any time, we now write its dynamics as the estimate $\hat{x}_t^{(i)}$. As we just described, this estimate is an exact solution up to the point where the first assembly $m$ to become fully activated does so at time $\bar{t}_m$, i.e. $\hat{x}_t^{(i)} = x_t^{(i)}$ for $i \le m$ and $t \le \bar{t}_m$. If these conditions are not met, we consider $\hat{x}_t^{(i)}$ to be an estimate of $x_t^{(i)}$. Note that, in general, $\hat{x}_t^{(i)} \ge x_t^{(i)}$.

**Solution for the first assembly.** We have already found the general solution for the first assembly before it becomes fully activated, given by Eq (41). From it we know that:

$$\hat{x}_t^{(1)} = \begin{cases} 0 & \text{for} \quad t = 0, \\ J \cdot \sum_{k=1}^{t} \left(\frac{R}{U}\right)^{k-1} & \text{for} \quad t \ge 1. \end{cases} \tag{51}$$

**Solution for the second assembly.** From Eq (51), we can see that

$$\hat{x}_t^{(1)} - \hat{x}_{t-1}^{(1)} = J \cdot \left(\frac{R}{U}\right)^{t-1}, \quad \forall \quad t \ge 1 \tag{52}$$

With this, we can compute from Eq (50) the solutions for the second assembly for $t \ge 2$:

$$\begin{aligned} \hat{x}_2^{(2)} &= J \cdot F/U \\ \hat{x}_3^{(2)} &= J \cdot (F/U) \cdot (1 + 2R/U) \\ \hat{x}_4^{(2)} &= J \cdot (F/U) \cdot (1 + 2R/U + 3(R/U)^2) \\ \hat{x}_5^{(2)} &= J \cdot (F/U) \cdot (1 + 2R/U + 3(R/U)^2 + 4(R/U)^3) \\ &\dots \end{aligned}$$

which in general can be written as

$$\hat{x}_t^{(2)} = \begin{cases} 0 & \text{for} \quad t < 2, \\ J \cdot \frac{F}{U} \cdot \sum_{k=2}^{t} (k-1) \left(\frac{R}{U}\right)^{k-2} & \text{for} \quad t \ge 2. \end{cases} \tag{53}$$

**Solution for the third assembly.** From Eq (53), we can see that

$$\hat{x}_t^{(2)} - \hat{x}_{t-1}^{(2)} = J(F/U)(t-1)(R/U)^{t-2}, \quad \forall \quad t \ge 2 \tag{54}$$

Thus, the solutions for the third assembly for $t \ge 3$ are:

$$\begin{aligned} \hat{x}_3^{(3)} &= J \cdot (F/U)^2 \\ \hat{x}_4^{(3)} &= J \cdot (F/U)^2 \cdot (1 + 3R/U) \end{aligned}$$

$$\hat{x}_5^{(3)} = J \cdot (F/U)^2 \cdot (1 + 3R/U + 6(R/U)^2)$$

$$\hat{x}_6^{(3)} = J \cdot (F/U)^2 \cdot (1 + 3R/U + 6(R/U)^2 + 10(R/U)^3)$$

$$\ldots$$

which in general can be written as

$$\hat{x}_t^{(3)} = \begin{cases} 0 & \text{for} \quad t < 3, \\ J \cdot \left(\dfrac{F}{U}\right)^2 \cdot \displaystyle\sum_{k=3}^{t} \binom{k-1}{2} \left(\dfrac{R}{U}\right)^{k-3} & \text{for} \quad t \geq 3. \end{cases} \tag{55}$$

where $\binom{n}{p}$ is the corresponding binomial coefficient.

**General solution.** If we continue this process iteratively, we will find that

$$\hat{x}_t^{(i)} = \begin{cases} 0 & \text{for} \quad t < i, \\ J \cdot \left(\dfrac{F}{U}\right)^{i-1} \cdot \displaystyle\sum_{k=i}^{t} \binom{k-1}{i-1} \left(\dfrac{R}{U}\right)^{k-i} & \text{for} \quad t \geq i \end{cases} \tag{56}$$

for $i \geq 1$ and $t \geq 1$, and initial conditions $\hat{x}_0^{(i)} = 0$.

**Proof of Eq** (56) **by induction.** We have just inferred a general expression for $\hat{x}_t^{(i)}$ by iterating Eq (50) from our initial conditions. Here, we will prove by induction that Eq (56) is indeed the solution to the difference equation Eq (50) for any $i \geq 1$ and $t \geq 1$, given initial conditions $\hat{x}_0^{(i)} = 0$.

Let us first consider the base case, $i = 1$. There, Eq (56) becomes

$$\hat{x}_t^{(1)} = \begin{cases} 0 & \text{for} \quad t < 1 \\ J \cdot \displaystyle\sum_{k=1}^{t} \left(\dfrac{R}{U}\right)^{k-1} & \text{for} \quad t \geq 1, \end{cases} \tag{57}$$

which is equivalent to Eq (41) that was shown to be the solution of $x_t^{(1)}$ while $0 \leq x_t^{(1)} \leq U$. Since $\hat{x}_t^{(1)} = x_t^{(1)}$ for $0 \leq x_t^{(1)} \leq U$, Eq (57) is indeed the solution to Eq (50) for the base case $i = 1$.

□

We will now do the induction step, assuming that the solution given by Eq (56) is true for some assembly $i = l - 1$, i.e.,

$$\hat{x}_t^{(l-1)} = \begin{cases} 0 & \text{for} \quad t < l - 1 \\ J \cdot \left(\dfrac{F}{U}\right)^{l-2} \cdot \displaystyle\sum_{k=l-1}^{t} \binom{k-1}{l-2} \left(\dfrac{R}{U}\right)^{k-(l-1)} & \text{for} \quad t \geq l - 1. \end{cases} \tag{58}$$

To derive $\hat{x}_t^{(l)}$, we use the difference equation Eq (50) for $i = l$, i.e.,

$$\hat{x}_t^{(l)} = \hat{x}_{t-1}^{(l)} + R \cdot \frac{\hat{x}_{t-1}^{(l)} - \hat{x}_{t-2}^{(l)}}{U} + F \cdot \frac{\hat{x}_{t-1}^{(l-1)} - \hat{x}_{t-2}^{(l-1)}}{U}. \tag{59}$$

We first consider $t < l$. In this case, we have $t-1 < l-1$, and therefore our induction hypothesis Eq (58) tells that $\hat{x}_{t-1}^{(l-1)} = \hat{x}_{t-2}^{(l-1)} = 0$. We can plug this into Eq (59) and find

$$\hat{x}_t^{(l)} = \hat{x}_{t-1}^{(l)} + R \cdot \frac{\hat{x}_{t-1}^{(l)} - \hat{x}_{t-2}^{(l)}}{U} \quad \text{for} \quad t < l. \tag{60}$$

For convenience, let us define the change in $\hat{x}_t^{(i)}$ in one time step:

$$\Delta \hat{x}_t^{(i)} := \begin{cases} 0 & \text{for} \quad t = 0, \\ \hat{x}_t^{(i)} - \hat{x}_{t-1}^{(i)} & \text{for} \quad t \geq 1 \end{cases} \tag{61}$$

such that we can write

$$\hat{x}_t^{(i)} = \sum_{k=0}^{t} \Delta \hat{x}_k^{(i)} \quad \text{for} \quad t \geq 0. \tag{62}$$

We can now write Eq (60) as

$$\Delta \hat{x}_t^{(l)} = \frac{R}{U} \cdot \Delta \hat{x}_{t-1}^{(l)} \quad \text{for} \quad t < l \tag{63}$$

which, given our initial condition $\Delta \hat{x}_0^{(l)} = 0$, has solution $\Delta \hat{x}_t^{(l)} = 0$ for any $t < l$ and, consequently, $\hat{x}_t^{(l)} = 0$ for any $t < l$. Thus, we have shown by induction that $\hat{x}_t^{(i)} = 0$ is true for any $i \geq 1$ and any $0 \leq t < i$. In other words, pulse propagation requires at least one assembly per time step; faster propagation is impossible.

$\square$

To complete our proof by induction, we will now consider also the case where $t \geq l$. The induction hypothesis given by Eq (58) for time $t-1$ is

$$\hat{x}_{t-1}^{(l-1)} = \begin{cases} 0 & \text{if} \quad t-1 < l-1, \\ J \cdot \left(\frac{F}{U}\right)^{l-2} \cdot \sum_{k=l-1}^{t-1} \binom{k-1}{l-2} \left(\frac{R}{U}\right)^{k-(l-1)} & \text{if} \quad t-1 \geq l-1, \end{cases} \tag{64}$$

and for time $t-2$ is

$$\hat{x}_{t-2}^{(l-1)} = \begin{cases} 0 & \text{if} \quad t-2 < l-1, \\ J \cdot \left(\frac{F}{U}\right)^{l-2} \cdot \sum_{k=l-1}^{t-2} \binom{k-1}{l-2} \left(\frac{R}{U}\right)^{k-(l-1)} & \text{if} \quad t-2 \geq l-1. \end{cases} \tag{65}$$

Thus, whether $t = l$ or $t > l$, the hypothesis in Eq (58) leads to

$$\hat{x}_{t-1}^{(l-1)} - \hat{x}_{t-2}^{(l-1)} = J \cdot \left(\frac{F}{U}\right)^{l-2} \cdot \binom{t-2}{l-2} \left(\frac{R}{U}\right)^{t-l} \quad \text{for} \quad t \geq l. \tag{66}$$

Plugging this into the difference equation Eq (59), we obtain

$$\hat{x}_t^{(l)} = \hat{x}_{t-1}^{(l)} + R \cdot \frac{\hat{x}_{t-1}^{(l)} - \hat{x}_{t-2}^{(l)}}{U} + J \cdot \left(\frac{F}{U}\right)^{l-1} \cdot \binom{t-2}{l-2} \left(\frac{R}{U}\right)^{t-l} \quad \text{for} \quad t \geq l \tag{67}$$

which we can write as

$$\Delta \hat{x}_t^{(l)} = \frac{R}{U} \cdot \Delta \hat{x}_{t-1}^{(l)} + J \cdot \left(\frac{F}{U}\right)^{l-1} \cdot \binom{t-2}{l-2}\left(\frac{R}{U}\right)^{t-l} \tag{68}$$

for $t \geq l$, where we used the definition in Eq (61). Then, we can find (proof in the following equation) that the explicit solution to Eq (68) is:

$$\Delta \hat{x}_t^{(l)} = J \cdot \left(\frac{F}{U}\right)^{l-1} \cdot \binom{t-1}{l-1}\left(\frac{R}{U}\right)^{t-l} \quad \text{for} \quad t \geq l. \tag{69}$$

To show that Eq (69) is indeed the solution to Eq (68), we can plug in $\Delta \hat{x}_{t-1}^{(l)}$ as given by Eq (69) into Eq (68) and obtain:

$$\Delta \hat{x}_t^{(l)} = J \cdot \left(\frac{F}{U}\right)^{l-1} \cdot \left[\binom{t-2}{l-1} + \binom{t-2}{l-2}\right]\left(\frac{R}{U}\right)^{t-l}$$

$$= J \cdot \left(\frac{F}{U}\right)^{l-1} \cdot \binom{t-1}{l-1}\left(\frac{R}{U}\right)^{t-l} \quad \square \tag{70}$$

To find $\hat{x}_t^{(l)}$ for $t \geq l$, we now only have to use Eq (62), which becomes

$$\hat{x}_t^{(l)} = \sum_{k=l}^{t} \Delta \hat{x}_k^{(l)}, \tag{71}$$

since we already showed that $\Delta \hat{x}_k^{(l)} = 0$ for $k < l$. Thus, we obtain

$$\hat{x}_t^{(l)} = J \cdot \left(\frac{F}{U}\right)^{l-1} \cdot \sum_{k=l}^{t} \binom{k-1}{l-1}\left(\frac{R}{U}\right)^{k-l}, \quad \text{for} \quad t \geq l \tag{72}$$

which completes our proof by induction.

$\square$

**Recovering the solution for $R = 0$.** Note that if $R = 0$, the solution given by Eq (56) becomes

$$\hat{x}_t^{(i)} = \begin{cases} 0 & \text{for} \quad t < i \\ J \cdot \left(\frac{F}{U}\right)^{i-1} & \text{for} \quad t \geq i, \end{cases} \tag{73}$$

which matches what we found in Eq (49) for a sequence without recurrent connections.

In summary, in Eq (56) we have derived a general solution for $\hat{x}_t^{(i)}$, with $\hat{x}_t^{(i)} = x_t^{(i)}$ for $i \leq m$ and $t \leq \bar{t}_m$ where assembly $m$ is the first assembly to become fully activated at time $\bar{t}_m$. Thus, we know that $m$ and $\bar{t}_m$ will correspond to the lowest $i$ and $t$ for which condition $\hat{x}_t^{(i)} \geq U$ is met. In terms of Eq (56), we can rephrase this as follows:

$$\frac{J}{U} \cdot \left(\frac{F}{U}\right)^{i-1} \sum_{k=i}^{t} \binom{k-1}{i-1}\left(\frac{R}{U}\right)^{k-i} \geq 1. \tag{74}$$

**Finding the pulse speed with the linear estimate.** If we assume Eq (56) is a good estimate of the model dynamics even if $i > m$ and $t > \bar{t}_m$, we can estimate the speed $S$ of successful replay pulses in general, by setting $t = \lfloor q/S \rfloor$ and $i = q$

in Eq (74), obtaining

$$\frac{J}{U} \cdot \left(\frac{F}{U}\right)^{q-1} \sum_{k=q}^{\lfloor q/S \rfloor} \binom{k-1}{q-1} \left(\frac{R}{U}\right)^{k-q} \geq 1 \,. \tag{75}$$

Since the left-hand side of Eq (75) increases monotonically as $S$ decreases, if the condition is met for a certain speed $S'$, it will also be met for all speeds $S < S'$. Thus, a pulse speed estimate $\hat{S}$ can be obtained by fixing $R$ and $F$ and finding the largest $S$ for which Eq (75) is true. If this condition is not met even for $S \to 0$, then replay is impossible.

**Comparing the linear estimate to rectangle-model simulations.** In Fig 4–E, we see that the analytical condition in Eq (75) becomes a better and better approximation of the model simulations as the sequence becomes longer. We saw above that the linear estimate $\hat{x}_t^{(i)}$ is exact for $i \leq m$ and $t \leq \bar{t}_m$ if $m$ is the first assembly to fully activate, and this happens at time $\bar{t}_m$. This is never the case in our simulations of Fig 4, where we start the pulse by fully activating the first assembly and determine the successful replay if there is a time for which the last assembly reaches $x_t^{(q)} \geq U$. For successful replays, all assemblies in the sequence are fully activated, so all assemblies $i < q$ can contribute to the error of the linear estimate $\hat{x}_t^{(q)}$. However, for failed replays, none of the other assemblies ($i > 1$) are fully activated. In these cases, the only error in $\hat{x}_t^{(q)}$ is caused by the first assembly. Thus, as the sequence length $q$ increases, this error becomes smaller and smaller. Since the analytical curves in Fig 4–E estimate the border between failed and successful replays, they become an exact match to the simulations of the rectangle model in the limit when $q \to \infty$.

**The limit of the linear estimate for infinitely long sequences.** Let us start by rearranging Eq (75) as

$$\frac{F}{U} \left[ \sum_{k=q}^{q/S} \binom{k-1}{q-1} \left(\frac{R}{U}\right)^{k-q} \right]^{1/q} \geq \left(\frac{F}{J}\right)^{1/q} \,. \tag{76}$$

Note that we have assumed $q/S$ to be an integer. To be strict, we could have explicitly written $\lfloor q/S \rfloor$, which denotes the integer value just under $q/S$. However, because we are interested in the limit $q \to \infty$, we assume $q/S \approx \lfloor q/S \rfloor$ for simplicity.

We first look at the right-hand side of Eq (76) and note that

$$\lim_{q \to \infty} \sqrt[q]{F/J} = 1 \,, \tag{77}$$

since $F$ and $J$ are positive constants. So, the limit of Eq (74) does not depend on $J$. Although the left-hand side of Eq (76) has a similar power $1/q$, its limit is less trivial to determine because the term inside the square brackets depends on $q$. Let us first define the function

$$\mathbb{Z}(x) := \sum_{k=q}^{q/S} \binom{k-1}{q-1} \cdot x^{k-q} \,. \tag{78}$$

To find the limit of Eq (76) we need to find

$$\lim_{q \to \infty} \left[ \mathbb{Z}\left(\frac{R}{U}\right) \right]^{1/q} \,. \tag{79}$$

We have defined $\mathbb{Z}$ as a sum of terms over the integer indices

$$\{k \in \mathbb{Z} \mid q \leq k \leq q/S\} \tag{80}$$

For convenience, we will now define $\mathcal{Z}$ in terms of a new non-integer index $m$ and write

$$\mathcal{Z}(x) = \sum_{m \in M} \binom{m \cdot q - 1}{q - 1} \cdot x^{(m-1)q}, \tag{81}$$

where

$$M = \left\{ \frac{k}{q} \mid k \in \mathbb{Z}, \, q \leq k \leq q/S \right\}. \tag{82}$$

Let us consider each term in the sum

$$T_m^{(q)}(x) := \binom{m \cdot q - 1}{q - 1} \cdot x^{(m-1)q}. \tag{83}$$

We can write this as

$$T_m^{(q)}(x) = \frac{(mq - 1)!}{(q - 1)! \cdot [(m - 1)q]!} \cdot x^{(m-1)q} \tag{84}$$

where we used the definition of the binomial coefficient:

$$\binom{n}{p} = \frac{n!}{p!\,(n - p)!}. \tag{85}$$

The factorials in the above expression can be written using Stirling's formula, i.e.,

$$n! = n^n e^{-n} P(\sqrt{n}), \quad \text{with} \quad P\left(\sqrt{n}\right) = \sqrt{2\pi n} \left[ 1 + \frac{1}{12n} + \mathcal{O}\left(\frac{1}{n^2}\right) \right]. \tag{86}$$

Note that $P\left(\sqrt{n}\right)$ is a polynomial that grows at most with $\sqrt{n}$. By doing this, we obtain

$$
\begin{aligned}
T_m^{(q)}(x) &= \frac{(mq - 1)^{mq-1}}{(q - 1)^{q-1}\,[(m - 1)q]^{(m-1)q}} \cdot \frac{P(\sqrt{mq - 1})}{P(\sqrt{q - 1}) \cdot P(\sqrt{(m - 1)q})} \cdot x^{(m-1)q} \\
&= \left[ \frac{(m - 1/q)^m}{1 - 1/q} \cdot \left( \frac{x}{m - 1} \right)^{m-1} \right]^q \cdot \tilde{P},
\end{aligned}
\tag{87}
$$

where in the second line we isolated the terms that grow exponentially with $q$, and represent the fraction of polynomials by the symbol $\tilde{P}$—note that these include not only the three polynomials in the first line but also an additional factor $q - 1/(mq - 1)$.

For convenience, we now define:

$$A_m^{(q)}(x) := \frac{(m - 1/q)^m}{1 - 1/q} \cdot \left( \frac{x}{m - 1} \right)^{m-1}. \tag{88}$$

and write Eq (79) as

$$\lim_{q \to \infty} \left[ \mathcal{Z}\left( \frac{R}{U} \right) \right]^{1/q} = \lim_{q \to \infty} \left( \sum_{m \in M} \left[ A_m^{(q)}(R/U) \right]^q \cdot \tilde{P} \right)^{1/q}. \tag{89}$$

If some of the terms $A_m^{(q)}$ are larger than 1, then as $q \to \infty$, $\mathcal{Z}(x)$ will be dominated by the term with the largest $A_m^{(q)}$. Thus, we now examine the conditions under which the terms $A_m^{(q)}$ are larger than 1. We know that $1 \le m \le 1/S$, so we first calculate

$$\lim_{m \to 1} A_m^{(q)}(x) = 1. \tag{90}$$

To determine if $A_m^{(q)}$ increases with increasing $m$, we calculate when the sign of its derivative with respect to $m$ is positive. Since $A_m^{(q)}$ is always non-negative, we can do this by looking at the simpler derivative of its logarithm:

$$\frac{\partial}{\partial m} A_m^{(q)}(x) > 0$$
$$\Leftrightarrow \quad \frac{\partial}{\partial m} \ln\left(A_m^{(q)}(x)\right) > 0$$
$$\Leftrightarrow \quad \ln\left(\frac{m - 1/q}{m - 1} \cdot x\right) + \frac{1}{mq - 1} > 0. \tag{91}$$

Since we are interested in the limit $q \to \infty$, we consider the second term on the left to be negligible and find that the condition for which the natural logarithm is positive is

$$\left(m < \frac{1 - x/q}{1 - x} \wedge x < 1\right) \vee \left(m > -\frac{1 - x/q}{x - 1} \wedge x > 1\right) \tag{92}$$

Here, we also consider $1 - x/q > 0$ because we are interested in large values of $q$. Thus, we find that, for $x < 1$, $A_m^{(q)}$ increases monotonously with increasing $m$ until a local maximum is reached at $(1 - x/q)/(1 - x)$. For $x > 1$, $A_m^{(q)}$ is always increasing. Thus, we have found that there are always terms $A_m^{(q)}$ that are larger than 1 and Eq (89) can be simplified:

$$\lim_{q \to \infty} \left[\mathcal{Z}\left(\frac{R}{U}\right)\right]^{1/q} \approx \max_{m \in M} \left\{m^m \cdot \left(\frac{R/U}{m - 1}\right)^{m-1}\right\}. \tag{93}$$

In this limit, the terms $1/q$ in Eq (87) have vanished, as have the other terms, since

$$\lim_{q \to \infty} \tilde{P}^{1/q} = 1. \tag{94}$$

Finally, to find the limit of Eq (74), we need to find the value of $m$ that maximizes

$$m^m \cdot \left(\frac{R/U}{m - 1}\right)^{m-1}. \tag{95}$$

From our analysis of the dependence of $A_m^{(q)}(x)$ on its parameters (see Eq (92) and its description), we know that if $x > 1$, the maximum value will be located at $m = 1/S$, which is the highest $m$ can be. For $x < 1$ and using $x = R/U$, the maximum is at $m = 1/(1 - R/U)$. However, this value might be outside the domain $1 \le m \le 1/S$. If that is the case, i.e.,

$$\frac{1}{1 - R/U} > 1/S \Leftrightarrow R/U > 1 - S \tag{96}$$

then the maximizing $m$ again is $1/S$. However, if

$$\frac{1}{1-R/U} \leq 1/S \Leftrightarrow R/U \leq 1-S \tag{97}$$

then the maximizing $m$ is $1/(1-R/U)$, and we have a local maximum. If we insert these two conditions, and respective maximizing values $m = 1/S$ or $m = 1/(1-R/U)$, into Eq (93) we get

$$\lim_{q\to\infty}\left[z\left(\frac{R}{U}\right)\right]^{1/q} \approx \begin{cases} \frac{1}{1-R/U} & \text{for} \quad R/U \leq 1-S \\ (1/S)\left(\frac{R/U}{1-S}\right)^{1/S-1} & \text{for} \quad R/U > 1-S \end{cases} \tag{98}$$

Plugging this into Eq (76), we finally find that the limit of Eq (74) when $q \to \infty$ is approximately

$$F/U \geq \begin{cases} 1-R/U & \text{for} \quad R/U \leq 1-S \\ S\left(\frac{1-S}{R/U}\right)^{1/S-1} & \text{for} \quad R/U > 1-S \end{cases} \tag{99}$$

Note that in the limit of infinitely slow propagation, i.e., $\lim_{S\to 0}$, the condition becomes

$$F/U \geq \begin{cases} 1-R/U & \text{for} \quad R/U \leq 1 \\ 0 & \text{for} \quad R/U > 1 \end{cases} \tag{100}$$

which can simply be written as

$$F + R \geq U \tag{101}$$

since both $R$ and $F$ are non-negative parameters. This expression is equivalent to what we called in the main text Condition 1 (for $x_0 = 0$), which is a necessary condition for replay for any distribution shape.

**A necessary condition for sequence retrieval.** Let us assume that some assembly $m > 1$ became fully activated at some time $\bar{t}_m$, such that $\sum_{j=0}^{\bar{t}_m} a_j^{(i-1)} = 1$. Following Eq (26) (for $x_0 = 0$), the position of the right edge of the subsequent assembly $m + 1$ at any time step $t > \bar{t}_m$ is given by

$$x_t^{(m+1)} = R \cdot \sum_{k=0}^{t-1} a_k^{(m+1)} + F. \tag{102}$$

If we assume that assembly $m + 1$ has not yet been fully activated, we can use Eq (36), such that Eq (102) becomes

$$\begin{aligned} x_t^{(m+1)} &= R \sum_{k=1}^{t-1} \frac{x_k^{(m+1)} - x_{k-1}^{(m+1)}}{U} + F \\ &= (R/U) \cdot \left(x_{t-1}^{(m+1)} - x_0\right) + F \\ &= F + (R/U) \cdot x_{t-1}^{(m+1)}, \quad \text{while} \quad x_t^{(m+1)} < U \end{aligned} \tag{103}$$

which is a first-order inhomogeneous difference equation.

If $R = U$, Eq (103) becomes

$$x_t^{(m+1)} = F + x_{t-1}^{(m+1)}, \tag{104}$$

where $x_t^{(m+1)}$ increases at a constant rate $F$ and can thus certainly cross $U$ within a finite number of time steps.

If $R \neq U$, we have to find the solution to the inhomogeneous difference equation. The general solution of the inhomogeneous difference equation Eq (103) is given by

$$x_t^{(m+1)} = x_t^{(m+1),h} + x_t^{(m+1),p} \tag{105}$$

where $x_t^{(m+1),h}$ is the general solution to the corresponding homogeneous equation

$$x_t^{(m+1),h} = \frac{R}{U} \cdot x_{t-1}^{(m+1),h} \tag{106}$$

and $x_t^{(m+1),p}$ is a particular solution to the inhomogeneous Eq (103). The general solution of the homogeneous equation is trivial and given by

$$x_t^{(m+1),h} = K \cdot \left(\frac{R}{U}\right)^t \tag{107}$$

where $K \in \mathbb{R}$ is some constant value. To find the particular solution, we try a constant (or fixed point) solution $x^{(m+1),p}$ that solves Eq (103), such that

$$x^{(m+1),p} = F + (R/U) \cdot x^{(m+1),p}$$

$$\Leftrightarrow x^{(m+1),p} = \frac{F}{1 - R/U}$$

Thus, we have found the general solution for Eq (103) when $R \neq U$, which is given by

$$x_t^{(m+1)} = \frac{F}{1 - R/U} + K \cdot \left(\frac{R}{U}\right)^t \tag{108}$$

To determine the value of the constant $K$, we would need to use the initial conditions. However, Eq (103) applies only for $t > \bar{t}_m$ and we cannot use the value of $x_{\bar{t}_m}^{(m+1)}$ as an initial condition since this is also unknown. Thus, we cannot find the exact value of $K$. However, we can determine whether it is a negative or positive constant, depending on the value of $R$. According to Eq (108),

$$x_t^{(m+1)} - x_{t-1}^{(m+1)} = K \cdot \left[\left(\frac{R}{U}\right)^t - \left(\frac{R}{U}\right)^{t-1}\right] = K \cdot \left(\frac{R}{U}\right)^{t-1} \cdot \left(\frac{R}{U} - 1\right) \tag{109}$$

We know that this quantity has to be non-negative since, by definition, $x_t^{(m+1)}$ can never decrease. Thus, if $R > U$, it follows that $K \geq 0$ must be true. We can write Eq (108) for $R > U$ as

$$x_t^{(m+1)} = \frac{F}{1 - R/U} + C_1 \cdot \left(\frac{R}{U}\right)^t, \quad C_1 \geq 0 \tag{110}$$

Then it is trivial to see that $x_t^{(m+1)}$ grows at an increasing rate and can reach $U$ in a finite number of time steps.

On the other hand, when $R < U$, Eq (109) tells us that for $x_t^{(m+1)}$ never to decrease, $K$ must be a non-positive constant. In that case, we can write Eq (108) as

$$x_t^{(m+1)} = \frac{F}{1 - R/U} - C_2 \cdot \left(\frac{R}{U}\right)^t, \quad C_2 \geq 0 \tag{111}$$

We thus see that, if $R < U$, $x_t^{(m+1)}$ is a monotonically increasing function that asymptotically tends to the limit $F/(1-R/U)$ as $t \to \infty$.

To summarize, if some assembly $m$ crosses the threshold at some time $\bar{t}_m$, assembly $m+1$ will become fully activated if $F/(1 - R/U) > U \Leftrightarrow F + R > U$, if given enough time steps (note that once $x_t^{(m+1)} > U$ the dynamics of Eq (103) no longer apply, but we can nevertheless know that at some point this will happen). Thus, the full activation of an assembly will lead to the sequential activation of the entire sequence if $R+F > U$, given enough time steps. The closer $R+F$ is to $U$, the more time steps are needed. If $F + R \le U$, the assembly will never become fully activated within a finite number of time steps.

## Supporting information

**S1 Appendix. Relation between pulse width and speed.** We perform a quantitative comparison between the pulse width and speed of successful replays, measured in the simulations shown in Fig 1–D and Fig 4–A,C.
(PDF)

**S2 Appendix. Spiking models with different excitation-inhibition ratios.** We simulate the 'Model 2' spiking network (sketched in Fig 1–A2) with different EI ratios and different times for balancing the network.
(PDF)

**S3 Appendix.  Minimal spiking model with instant synapses.** We simulate our minimal spiking network (sketched in Fig 1–A3) with excitatory synapses that act as delta-current pulses.
(PDF)

## Acknowledgments

We thank Stefano Masserini, Natalie Schieferstein, and John Rinzel for very helpful discussions and feedback on the manuscript.

## Author contributions

**Conceptualization:** Gaspar Cano, Richard Kempter.

**Formal analysis:** Gaspar Cano, Richard Kempter.

**Funding acquisition:** Richard Kempter.

**Investigation:** Gaspar Cano.

**Methodology:** Gaspar Cano, Richard Kempter.

**Project administration:** Richard Kempter.

**Resources:** Richard Kempter.

**Software:** Gaspar Cano.

**Supervision:** Richard Kempter.

**Validation:** Gaspar Cano, Richard Kempter.

**Visualization:** Gaspar Cano.

**Writing – original draft:** Gaspar Cano.

**Writing – review & editing:** Gaspar Cano, Richard Kempter.

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
