## [Decision Letter · Decision Letter 0]

17 Jun 2025

PCOMPBIOL-D-25-00889

Conditions for replay of neuronal assemblies

PLOS Computational Biology

Dear Dr. Kempter,

Thank you for submitting your manuscript to PLOS Computational Biology. After careful consideration, we feel that it has merit but does not fully meet PLOS Computational Biology's publication criteria as it currently stands. Therefore, we invite you to submit a revised version of the manuscript that addresses the points raised during the review process.

Please submit your revised manuscript within 60 days Aug 17 2025 11:59PM. If you will need more time than this to complete your revisions, please reply to this message or contact the journal office at ploscompbiol@plos.org. Please include the following items when submitting your revised manuscript:

We look forward to receiving your revised manuscript.

Kind regards,

Jian Liu

Academic Editor

PLOS Computational Biology

Marieke van Vugt

Section Editor

PLOS Computational Biology

**Additional Editor Comments:**

The reviewers generally appreciate the work presented in the manuscript. However, addressing certain concerns raised by the reviewers could further improve the clarity and overall quality of the paper.

**Journal Requirements:**

At this stage, the following Authors/Authors require contributions: Gaspar Cano, and Richard Kempter. Please ensure that the full contributions of each author are acknowledged in the "Add/Edit/Remove Authors" section of our submission form.

5) We notice that your supplementary Figures are included in the manuscript file. Please remove them and upload them with the file type 'Supporting Information'. Please ensure that each Supporting Information file has a legend listed in the manuscript after the references list.

**Reviewers' comments:**

Reviewer's Responses to Questions

**Comments to the Authors:**

Reviewer #1: In this paper, the authors investigate the fundamental conditions for groups of neurons to fire simultaneously in precise sequences, known as synfire chains. In particular, the authors rather than emphasizing the significant role of inhibitory feedback, demonstrated and highlighted the critical importance of the interply between inter-assembly feedforward connectivity and the intra-assembly recurrent connectivity. These structural motifs (at multiple scales and hierarchies) are shown to be key factors for amplifying and sustaining neuronal synfire chain propagation.

To support their observations and conclusions, the authors moreover developed an analytical framework meanwhile constructed minimal SNN models. Their combined analytical and numerical results convincingly demonstrated that recurrent connections alone can support the generation and replay of sequential activity.

This work contributes to our understanding of the mechanisms underlying synfire chains and sequence replay, with potential implications for theoreis of memory consolidation and sequential processing.

Overall, I think this work to be of high relevance and suitable for Plos CB, however, several major revisions and clarifications are necessary before it can be considered for publication.

Major:

1. The paper appears to be organized into three main parts: it begins with a behavioral analysis and comparison of sequence retrieval across different SNN architectures, then introduces a theoretical population model to quantify key aspects of sequence retrieval and identify three essential conditions for successful replay, and finally returns to the minimal spiking models to validate the theoretical predictions through numerical simulations.

In the first part, such as Fig. 1 and related results, the authors seem to focus on the role of E-I balance by comparing three SNN architectures, then demonstrating (emphasizing?) that inhibitory feedback is not a necessary condition for sequence replay. In the second part, however, the theoretical framework seems to be developed specifically based on the minimal SNN (Model 3), which only includes excitatory neurons. Here, the emphasis shifts from inhibition to the interaction between feedforward (F) and recurrent (R) connectivity. This analysis leads to the identification of three critical conditions for sequence replay, formulated as functions of F and R. In the final section, the study returns to the minimal SNN to numerically validate the influence of F-R interaction on sequence replay.

This article flow is good, but I believe a reorganization could improve the clarity and narrative logic of the manuscript. For instance, the authors might consider restructuring the presentation to first introduce the three SNN architectures to show that inhibitory feedback is not essential. Then, focusing solely on the minimal model (model three), they could deeply explore how varying network parameters affects sequence retrieval, as done in current Fig. 4. This exploration would naturally motivate the development of the theoretical framework, which would then serve to explain and generalize the key observations, culminating in the derivation of the three critical conditions for successful replay.

Of course, the authors may have an alternative structure in mind that best fits their goals, and I leave the decision to them. I would moreover recommend that the author put less emphasis on the unnecessary of the inhibitory feedback.

2. The limitation of the current population model appears to be missing from being discussed. For example, if I understand correctly, the model resembles a rate-based or mean-field approximation, relying on the assumption that individual neurons within an assembly activate independently. However, this assumption may break down when recurrent excitation (e.g., W^{EE}) becomes sufficiently strong. In such regimes, the activation of a single neuron could trigger a cascade of activity within the assembly, resulting in a so-called "neural avalanche" (see for example https://journals.aps.org/prl/abstract/10.1103/PhysRevLett.134.028401). This phenomenon may correspond to the “explosion” described in the manuscript.

I understood that the authors have taken care to embed their analysis within an appropriate parameter regime, it would be better to explicitly state the limitations of the population model and clarify the valid operating regime in which its assumptions hold.

And the authors state that the population model provides a better approximation as the sequence length q increases. A related question is whether the model also implicitly assumes a sufficiently large number of neurons within each assembly? I would expect that having a large enough assembly size is necessary to ensure that the statistical assumptions of the voltage distribution remain valid.

3. The authors emphasize that recurrent and feedforward connectivity work in concert to amplify neuronal activity and facilitate sequence retrieval. There is also a complementary line of research that explores the competition between these two forms of connectivity. These studies highlight their respective roles in balancing and regulating information transmission versus information processing and computation, for example, Clark, D. G., & Beiran, M. (2025) PNAS; Wang, Z., Sornborger, A. T., & Tao, L. (2016). PlosCB. It would strengthen the manuscript if the authors could briefly discuss this perspective in the discussion section, particularly in the broader context of how recurrent and feedforward pathways jointly shape information flow in the brain.

Minor:

1. Some subsubsections could be merged for improved coherence and flow. For example, the sections “Relation between pulse speed and connectivity” and “Necessary connectivity given a minimum speed” both address the relationship between sequence speed and connectivity structure and could be merged.

2. The line following S3. 22, a typo "with".

Is the second line in S3.58 missing a (q-1)/(mq-1) term?

Reviewer #2: The paper examines neuronal sequences through synfire chains to simulate hippocampal replay. Interestingly, it finds that feedback inhibition is not required for the formation of firing sequences. Additionally, the paper provides an analytical description of how various connectivity structures influence speed, indicating that weaker feedforward connections produce slower pulses, which can be maintained by recurrent connections. Below are several suggestions and queries that we hope will strengthen the paper.

1. (Major) The model establishes a fixed ratio of E- to I- cells at 4:1. However, Gandolfi et al. (2023) reconstructed the human CA1 region using a 9:1 cell-type ratio. Would an E:I ratio greater than 4 influence your key analysis on "conditions for replay" or the stability of sequence propagation? If so, please discuss the sensitivity of the parameters involved and the physiological rationale for selecting a 4:1 ratio.

Reference: Gandolfi, D., Mapelli, J., Solinas, S.M.G. et al. Full-scale scaffold model of the human hippocampus CA1 area. Nat Comput Sci 3, 264–276 (2023). https://doi.org/10.1038/s43588-023-00417-2

2. (Major) The model fixes the number of excitatory assemblies at q = 10 for insets A and C of Figure 3. How about other numbers of assemblies, say, q = 200?

3. (Major) The current model applies STDP mechanisms only at I to E synapses to homeostatically regulate EI balance without altering the fixed feedforward chain. It would be valuable to discuss what happens if STDP were also applied to E-E, E-I, or I-I connections where these STDPs have existed in real animal experiments. For example, E-E plasticity might enable the assembly chain to be learned or refined, but could drift away from the derived replay conditions unless counterbalanced by homeostatic mechanisms. Similarly, plasticity at E-I or I-I synapses could modify pulse width and propagation speed. A brief discussion or supplemental simulations exploring these scenarios would clarify the robustness of the replay conditions.

4. (Major) I wonder about the situation if there is a model 4 that has background connections but only uses Poisson neurons as inhibitory assemblies?

5. (Major) In lines 100-103, the authors discuss activity pulse propagation. I wonder if there is empirical criteria or specific firing-pattern benchmarks (e.g., propagation delay, spike-time synchrony) they would use to determine that the modeled propagation accurately reflects real anatomical data.

6. (Major) In lines 146–147 the authors identify p_f = 0.12 as a threshold for stable propagation. Please clarify whether this critical feed-forward probability depends on network scale—specifically total excitatory neuron count (N_E) or assembly number (q)—and if so, how p_f shifts when these parameters vary.

7. (Major) The manuscript does not explain why it chose N_E = 5000 in model 3 but 20000 in models 1 and 2.

8. (Major) In Equation 19, the analysis focuses solely on the W_EE. Please clarify why other synaptic weights (e.g., W_EI, W_IE, W_II) were not included in the propagation criteria. How would varying those other weights affect the derived conditions for replay and the stability of sequence propagation?

9. (Minor) The manuscript contains many parameters, which the authors have organized into Table 1. However, this arrangement may make it difficult for readers to easily reference. They need go back and forth to read this Table. I recommend that the authors include the specific value for each parameter immediately for improved readability. This change would maintain the table's formal structure while enhancing clarity.

10. (Minor) For further clarity, each horizontal row in Figure 1B and 1C corresponds to one assembly (IDs 1-10) with the bottom-to-top rows, representing Assembly 1, the new row Assembly 2, and so on.

11. (Minor) The Figure 1 inset A3 missed a "N_E" beside the N_E assemblies.

12. (Minor) Figure 2 has unlabeled x-axes and missing arrows on the x-axes.

13. (Minor) The notation g_t(v) (Eq. 1) for the membrane-potential distribution may be mistaken for conductance. Consider using other notation to avoid confusion. Moreover, h_U(v) is a normalized function (in line 286), while it is a rectangle model in line 326. Please consider different notations as well.

14. (Minor) In Figure 3 panels A and C, please specify the heat maps plot F and R values in units.

15. (Suggestion) Please verify whether some corrections will affect the results.

Reviewer #3: This manuscript investigates how recurrent connections can support the replay of sequential neural activity in LIF spiking networks, even in the absence of inhibition or balanced amplification. The authors use a combination of simulations and a time-discrete population model to characterize conditions under which sequence retrieval (or replay) occurs and to quantify how replay speed and pulse width depend on network connectivity parameters. The study provides mechanistic insight into how excitatory networks alone can generate propagating activity patterns, a scenario relevant to cortical and hippocampal function. The manuscript addresses an important question in theoretical neuroscience and proposes a simplified yet analytically tractable framework. The findings challenge the conventional emphasis on inhibitory mechanisms in sequence replay, offering a conceptually original perspective. The use of a population-level membrane potential distribution model is a notable methodological contribution. However, the manuscript suffers from issues in organization, clarity, and presentation. The argumentation is fragmented, the narrative lacks coherence, and key results are often buried in a sea of technical detail and assumptions. A thorough reorganization is needed to improve readability and highlight the main scientific contributions.

Strengths:

1. Introduces a tractable population model using membrane potential distributions.

2. Derives analytical conditions predicting replay feasibility and speed.

3. Provides a mechanistic link between subthreshold membrane potential distributions and replay dynamics.

4. Proposes a minimal model that may better reflect local hippocampal/cortical microcircuits than previously assumed balanced EI models.

5. Exploration of pulse dynamics (speed, width) as functions of connectivity and synaptic time scales.

However, the manuscript is overly dense, with many small technical arguments that obscure the main ideas. The progression from spiking model results to the population model is unclear; better scaffolding is needed to guide the reader. Model assumptions are central to the population model but are presented in a scattered fashion. Their rationale, implications, and limitations should be further clarified. Equation (1) may be corrected as h_U(v) = g_t^(i) (v-x_t^(i)) in my understanding. The main insights (e.g., recurrent excitation alone can support replay; speed–width–connectivity tradeoffs) are not clearly foregrounded. The claim that leak currents are essential in determining subthreshold distribution is made but not explicitly supported by results — adding a dedicated figure or example could help substantiate this. Highlighting the transition from realistic spiking behavior to idealized population-level dynamics would clarify the model's scope.

**Have the authors made all data and (if applicable) computational code underlying the findings in their manuscript fully available?**

Reviewer #1: Yes

Reviewer #2: Yes

Reviewer #3: Yes

PLOS authors have the option to publish the peer review history of their article (what does this mean?). If published, this will include your full peer review and any attached files.

Reviewer #1: No

Reviewer #2: No

Reviewer #3: **Yes: **Dongping Yang

**Figure resubmission:**
---

## [Decision Letter · Decision Letter 1]

15 Dec 2025

Dear Dr. Kempter,

We are pleased to inform you that your manuscript 'Conditions for replay of neuronal assemblies' has been provisionally accepted for publication in PLOS Computational Biology.

Best regards,

Jian Liu

Academic Editor

PLOS Computational Biology

Marieke van Vugt

Section Editor

PLOS Computational Biology

This revised manuscript demonstrates substantial improvement and is nearly ready for publication. Please incorporate the final minor revisions suggested below to polish the presentation and finalize the submission.

Reviewer's Responses to Questions

**Comments to the Authors:**

Reviewer #1: The authors have addressed all of my questions and in particular have reorganized the manuscript to improve its clarity and flow. The paper is now suitable for publication in Plos CB. Congratulations.

Reviewer #2: Please see the attached PDF file.

Reviewer #3: The revised version of this paper has been significantly improved. I have no more concerns

**Have the authors made all data and (if applicable) computational code underlying the findings in their manuscript fully available?**

Reviewer #1: Yes

Reviewer #2: Yes

Reviewer #3: Yes

PLOS authors have the option to publish the peer review history of their article (what does this mean?). If published, this will include your full peer review and any attached files.

Reviewer #1: **Yes: **Yuxiu SHAO

Reviewer #2: No

Reviewer #3: **Yes: **Dongping Yang

---

## [Editor Report · Acceptance letter]

PCOMPBIOL-D-25-00889R1

Conditions for replay of neuronal assemblies

Dear Dr Kempter,

I am pleased to inform you that your manuscript has been formally accepted for publication in PLOS Computational Biology. Your manuscript is now with our production department and you will be notified of the publication date in due course.

With kind regards,

Zsofia Freund
